

# High-Resolution Analysis of the Physicochemical Characteristics of Sandstone Media at the Lithofacies Scale

Adrian Linsel[1], Sebastian Wiesler[1], Jens Hornung[1], and Matthias Hinderer[1]

[1]Technische Universität Darmstadt, Schnittspahnstr. 9, 64287 Darmstadt, Germany

**Correspondence:** Adrian Linsel (linsel@geo.tu-darmstadt.de)

**Abstract.** The prediction of physicochemical rock properties in subsurface models regularly suffers from uncertainty observed at the sub-meter scale. Although at this scale – which is commonly termed the lithofacies scale – the physicochemical variability plays a critical role for various types of subsurface utilization, its dependence on syn- and post-depositional processes is still subject to investigation.

The impact of syn- and post-depositional geological processes, including depositional dynamics, diagenetical compaction and chemical mass transfer, onto the spatial distribution of physicochemical properties in siliciclastic media at the lithofacies scale is investigated in this study. We propose a new workflow using two cubic rock samples where eight representative geochemical, thermophysical, elastic and hydraulic properties are measured on the cubes' faces and on samples taken from the inside. The scalar fields of the properties are then constructed by means of spatial interpolation. The rock cubes represent
the structurally most homogeneous and most heterogeneous lithofacies types observed in a Permian lacustrine delta formation that deposited in an intermontane basin. The spatio-temporal controlling factors are identified by exploratory data analysis and geostatistical modeling in combination with thin section and environmental scanning electron microscopy analyses.

Sedimentary structures are well preserved in the spatial patterns of the negatively correlated permeability and mass fraction of $Fe_2O_3$. The Fe-rich mud fraction, which builds large amounts of the intergranular rock matrix and of the pseudomatrix,
has a degrading effect onto the hydraulic properties. This relationship is underlined by a zonal anisotropy that is connected to the observed stratification. Feldspar alteration produced secondary pore space that is filled with authigenic products including illite, kaolinite and opaque phases. The local enrichment of clay minerals implies a non-pervasive alteration process that is expressed by network-like spatial patterns of the positively correlated mass fractions of $Al_2O_3$ and $K_2O$. Those patterns are spatially decoupled from primary sedimentary structures. The elastic properties, namely p- and s-wave velocity, indicate a
weak anisotropy that is not inevitably oriented perpendicularly to the sedimentary structures.

The multifarious patterns observed in this study emphasize the importance of high-resolution sampling in order to properly model the variability present in a lithofacies-scale system. In fact, the physicochemical variability observed at the lithofacies scale might nearly cover the global variability in a formation. Hence, if the local variability is not considered in full-field projects – where the sampling density is usually low – statistical correlations and thus conclusions about causal relationships
among physicochemical properties might be feigned inadvertently.





## 1 Introduction

The utilization of the subsurface in disciplines such as petroleum reservoir engineering, geothermal heat extraction, mining, carbon-capture and storage or nuclear waste disposal requires highly accurate spatial predictions of relevant physical or geochemical properties in order to assess the economic feasibility of a target region (Landa and Strebelle, 2002; Heap et al., 2017; Kushnir et al., 2018; Rodrigo-Ilarri et al., 2017). Although most of these types of utilization take place at full-field scales, geological variability present at the sub-meter scale may play an important role during the development process. The scale we are speaking of is commonly termed the lithofacies scale (Miall, 1985). Geological heterogeneities at the lithofacies scale might constitute undesirable features in the subsurface such as flow-barriers in reservoirs (Landa and Strebelle, 2002; Ringrose et al., 1993), pathways in nuclear waste disposal sites or geochemical anomalies in mining areas (Wang and Zuo, 2018). Hence, the controlling factors of sub-meter variability should be understood and at least roughly quantified before starting the development in the subsurface region.

In sedimentary bodies, the spatial distribution of the properties is mainly controlled by depositional and diagenetical processes (McKinley et al., 2011, 2013). The spatial characteristics of physicochemical properties in sedimentary rock media are complex due to strongly intersecting and interacting processes during sediment transport, deposition and diagenesis (McKinley et al., 2011). Multiple studies aimed to quantify the variability at the lithofacies scale, most of which concentrated on reservoir properties such as permeability and porosity in 2-D space (McKinley et al., 2011; Hornung et al., 2019). A 2-D analysis suits well for identifying non-visible patterns related to micro-bedding structures at multiple scales even in very homogeneous sandstones (McKinley et al., 2004). That perspective, however, involves simplifications of the physicochemical variability in 3-D space since specific rock properties such as permeability are dependent on the Cartesian direction. Also, consideration of geostatistical parameters such as variographic direction, range, sill and nugget revealed differences in 3-D compared to 2-D space (Landa and Strebelle, 2002; Hurst and Rosvoll, 1991).

With a proper knowledge of the statistical and causal relationships among physicochemical rock properties at different scales, prognostic property models can be significantly enhanced by the integration of small-scale uncertainty into upscaling or conditional simulation algorithms (Lake and Srinivasan, 2004; Verly, 1993). Especially, since multivariate geostatistics can account for interrelationships between rock properties, those relationships can be used as trends or drifts in geostatistical predictions in order to optimize their accuracy in space and time (Hudson and Wackernagel, 1994).

In order to quantify the spatial variability and the multidimensional relationships among physicochemical properties at the 3-D lithofacies scale, the quasi-continuous scalar fields of two rock cubes are modeled by means of spatial interpolation, which is constrained by laboratory measurements. The rock cubes have volumes of 0.0156 $m^3$ and 0.008 $m^3$ and are taken from a Permian lacustrine-deltaic sandstone formation that deposited in the intermontane Saar-Nahe basin during the Cisuralian series. The lithological characteristics of the sandstones are analyzed and both isotropic and anisotropic properties, including bulk rock geochemistry, thermophysical, hydraulic and elastic rock properties, are measured on the cubes' faces. In addition, the intrinsic gas permeability under infinite pressure gradient, the effective porosity, the elemental composition, the thermal





conductivity, the thermal diffusivity together with the p- and s-wave velocity are measured on 108 rock cylinders taken from
the inside of the cubes representative for each Cartesian direction in order to account for anisotropic patterns.

The measurements are used to interpolate the full 3-D field of each property. Prior to interpolation, the discrete measurements are statistically analyzed for correlation and formal relationships. Interpolations are conducted using deterministic and stochastic methods including the inverse distance weighting (IDW) and simple kriging (SK) interpolation. The models are evaluated through cross-validation and the observed spatial patterns are categorized. The interpolation results providing the lowest
cross-validation error are statistically analyzed again and compared with the aforementioned statistical patterns. Eventually, the geological processes, which produced the observed patterns, are interpreted and discussed with the help of qualitative thin section and environmental scanning electron microscope (ESEM) analyses.

## 2  Methodology

### 2.1  Sedimentological characterization and rock sampling

In order to cover multiple varieties of sedimentary lithofacies types, a quarry in Obersulzbach (Rhineland-Palatinate, Germany) in the Saar-Nahe basin was selected for the investigations (Fig. 1). The quarry belongs to the lacustrine-deltaic Disibodenberg formation that is assigned to the Innervariscan Rotliegend Group and comprises four lithofacies types. This formation is deeply buried (1,995 to 2,380 m b.g.s.) in the northern Upper Rhine Graben in southwestern Germany (Becker et al., 2012) and constitutes a potential target unit for hydrothermal exploitation (Aretz et al., 2015). The outcrop has been chosen in order
to estimate the variability of physicochemical properties that could be expected in this formation as an uncertainty factor if the formation gets targeted in a deep geothermal project.

Two rock cubes of $0.2 \times 0.2 \times 0.2$ m (OSB2_c) and $0.25 \times 0.25 \times 0.25$ m (OSB1_c) were taken from the outcrop wall using a rock chainsaw. According to the outcrop's coordinate system, one edge of the cuboid runs east-west (x), one north-south (y) and one in altitude (z) direction. The irregular cuboids were reworked to regular cubes with a stationary rock saw. We selected
two types of lithofacies (Fig. 2) – both sandstones – one representing a cross-bedded, heterogeneous, compartmentalized variety (OSB1_c) and the other one a homogeneous variety (OSB2_c). The cubes both were extracted from a distributary mouth bar building a foreset in a fluviatile-dominated lacustrine delta. OSB1_c (Fig. 2) was taken from the high-energetic basal part whereas OSB2_c was taken from the lower-energetic top. The sedimentological characteristics including grain size, sorting, granularity, sedimentary structures and mineral content were determined by visual inspection, thin section and ESEM
analyses. Two different types of zonal anisotropy and spatial patterns were expected to be found with the aforementioned sampling strategy. In other studies such as McKinley et al. (2011) measurements were directly conducted in the field. These approaches, however, do often provide a drawback in accuracy and precision, especially in permeability measurements. In order to address this issue, we measured the faces of the cubes under laboratory conditions. In the next step, the cubes were cut to rock slabs, from which cylinder samples were taken. Totally, 108 rock cylinders – 79 from OSB1_c and 29 from OSB2_c –





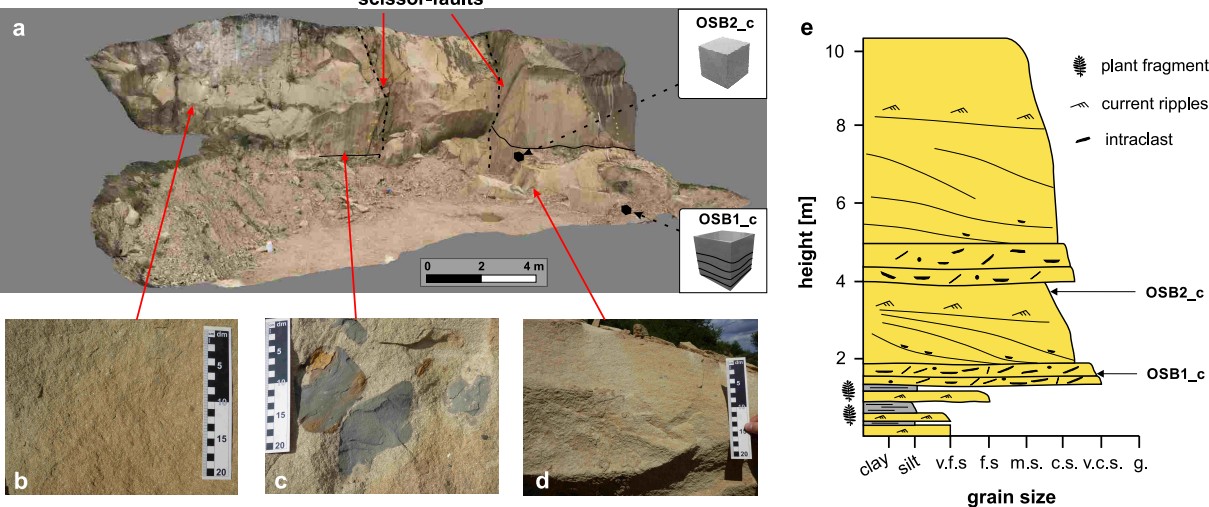

**Figure 1.** (a) The investigated sandstone quarry in Obersulzbach, Germany. The outcrop is compartmentalized by two scissor-faults in the central part that provide offsets of several meters. (b) Homogeneous sandstone (c) Rip-up clasts embedded in a homogeneous rock matrix (d) Ripple-cross bedded sandstone (e) Cumulative sedimentary log of the outcrop architecture. The sampling positions of OSB1_c and OSB2_c are marked. v.f.s. = very fine sand; f.s. = fine sand; m.s. = medium sand; c.s. = course sand; v.c.s. = very coarse sand; g. = granule.

were extracted from the rock cubes. It was ensured that at least five samples were produced representative for each Cartesian direction. Applying the formula for calculating a cylinder's volume $V_c$ with

$$V_c = h \times \pi \times r^2, \tag{1}$$

where $h$ is the height of the cylinder and $r$ the radius, the relative volume covered by the rock cylinders in the rock cubes was calculated to be 25.4% for OSB1_c and 18.2% for OSB2_c, respectively. Eventually, target meshes are needed to interpolate

the full 3-D scalar fields. Therefore, both cubes were modeled in 3-D using a regular grid consisting of 27,000 hexahedral, orthogonal cells.

## 2.2 Measurement campaign

First, a local metric coordinate system was defined, where each edge of the cube represents an axis in the Cartesian coordinate system in order to reference each measurement to a point in space. The sampling points were set in a raster of $9 \times 9$ points on

each face for OSB1_c and $5 \times 5$ for each face of OSB2_c. All measurements were conducted in the laboratory of the Institute of Applied Geosciences in Darmstadt, Germany. After drying the rock cubes at 60°C, non-invasive measurements were conducted on each face of the cube. On the cubes' faces the p- and s-wave velocity and elemental mass fractions were determined (Fig. 3).





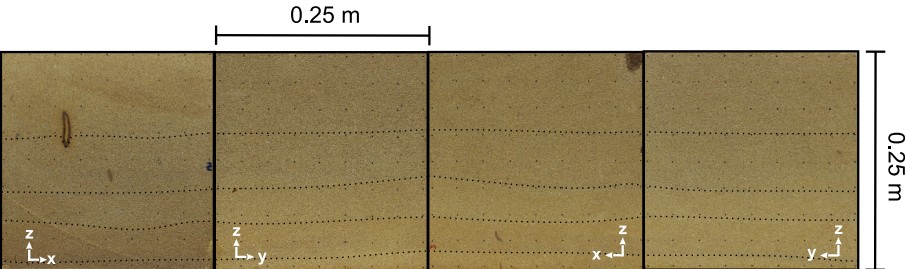

**Figure 2.** Lateral faces of OSB1_c displayed in the form of an open cube (from left to right: XZ Front, YZ Front, XZ Back and YZ Back). The internal bounding surfaces are indicated by the dashes lines.

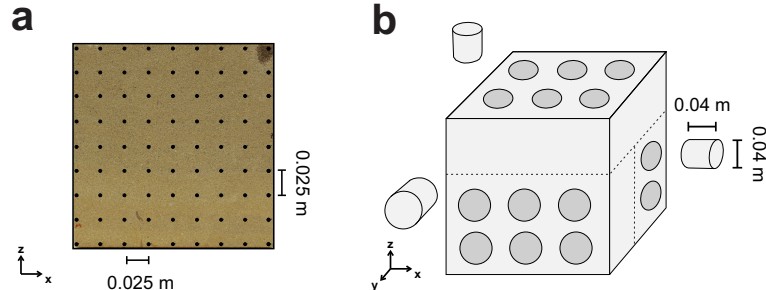

**Figure 3.** (a) Sampling locations for the non-invasive measurements including p- and s-wave velocity and x-ray fluorescence exemplary displayed for the face XZ Back of OSB2_c (b) Schematic of the extraction strategy for sampling the rock cylinders.

After the extraction, the rock cylinders were oven-dried at 105 °C and measured in order to determine the intrinsic gas
permeability, effective porosity, p- and s-wave velocity, elemental mass fractions, thermal conductivity as well as the thermal
diffusivity in unsaturated conditions. Those properties can be considered key properties of the rock matrix in porous aquifers
with regard to hydrothermal systems (Agemar et al., 2014) since they constitute input variables for the governing equations for
heat transfer and fluid flow in the subsurface (Carslaw and Jaeger, 1959).

The permeability was measured with the Hassler cell permeameter, which is described in Filomena et al. (2014). This tech-
nique allows for the estimation of the intrinsic gas permeability, which is the permeability at an infinite pressure gradient. The
Hassler cell permeameter was set to accept a measurement if fifteen consecutive readings do not deviate by more than 5%. The
measurement error, however, can exceed that value especially in low-permeable lithologies. Effective porosity measurements
were conducted using an envelope density analyzer (GeoPyc 1360). The accuracy is given by the manufacturer to be within $\pm$
0.55% (Micromeritics, 1998). Thermal properties under unsaturated conditions, namely the thermal conductivity and thermal
diffusivity, were determined with a thermal conductivity scanner (TCS) according to the work of Popov et al. (1999). The
measurement error is quantified to be $\leq 3\%$ for thermal conductivity and $\leq 8\%$ for thermal diffusivity (Popov et al., 1999).
The elastic properties of p- and s-wave velocity in the rock media were measured with the sonic wave generator UKS-D
(Geotron) by sending a sonic wave pulse from a pulse-providing test head (UPG-S) to a receiver (UPG-E). The wave velocity





is a function of travel length, density and travel time. The initial occurrence of the p- or s-wave must be picked manually after visual inspection by the operator. Thus no measurement error can be provided since user bias cannot be assessed quantitatively. Bulk elemental analysis using the Bruker S1 TITAN handheld portable X-ray fluorescent (pXRF) analyzer was used to find correlations between the elemental composition and the petrophysical properties. The measurement device works on the basis of energy dispersive X-ray fluorescence (EDXRF) and estimates the elemental mass fractions of a sample. This device produces an ionizing X-ray beam of 1.2 cm diameter and quantifies the elemental composition based on the energy emitted by the ionized elements in the targeted area. The portable device can measure the fraction of elements with an ordinal number $\geq 12$ and $\leq 235$ if the threshold value defined by the measurement error for the specific element in the sample is exceeded. For this study, the device was operated in GeoChem, DualMining mode allowing for detection of the major oxides $SiO_2$, $Al_2O_3$, $Fe_2O_3$ and $K_2O$ as well as a wide range of other elements. The device has been calibrated with international standards. We used the previously mentioned major oxides for analyses since those can provide insight into the iron oxide and clay mineral distribution, which can impact petrophysical properties significantly. More details on the measurement devices can be found in the works of Hornung and Aigner (2002), Sass and Götz (2012), Filomena et al. (2014) and Aretz et al. (2015).

### 2.3 Data analysis and spatial modeling

#### 2.3.1 Variography

The experimental semivariogram represents the cumulative dissimilarity of a discrete set of point-pairs $x$ with $n_c$ as the count of point-pairs within the distance classes $\boldsymbol{h}$ of identical distance increments (Eq. 2).

$$\gamma(\boldsymbol{h}) = \frac{1}{2n_c} \sum_{\alpha=1}^{n_c} \left( z(x_\alpha + \boldsymbol{h}) - z(x_\alpha) \right)^2 \tag{2}$$

The continuous counterpart, represented by the variogram model, is an approximation to the experimental semivariogram assuming $z(\boldsymbol{x})$ to be a stationary random field (Wackernagel, 2003). A variogram model $\gamma_{theo}$ is represented by a covariance function $c$ with the relationship $\gamma_{theo}(\boldsymbol{h}) = c(0) - c(\boldsymbol{h})$, where $c$ is a positive definite, even function. Six covariance models are mostly used to fit the experimental semivariogram, namely the spherical, gaussian, exponential, power, kardinal sine and the linear model (Armstrong, 1998; Ringrose and Bentley, 2015). In this study, we only observe spherical relationships with nugget effect. This model is calculated as

$$c_{sph}(\boldsymbol{h}) = \begin{matrix} n + b \cdot \left(1 - \frac{3|\boldsymbol{h}|}{2a} + \frac{|\boldsymbol{h}|^3}{2a^3}\right) & \text{for} & 0 \leq |\boldsymbol{h}| < a \\ n & \text{for} & |\boldsymbol{h}| \geq a, \end{matrix} \tag{3}$$

with the variables nugget (n), range (a) and sill (b). Semivariograms can be used to quantify the spatial or time correlation of a random property (Ringrose and Bentley, 2015; Gu et al., 2017; Rühaak et al., 2015). Further on, the differences in range and sill in dissimilar directional semivariograms can quantify the zonal and geometric anisotropy of a property (Ringrose and Bentley, 2015). The resulting covariance function is an input variable for stochastic interpolation algorithms.





### 2.3.2 Rock property interpolation

Spatial inter- and extrapolation can be generated with deterministic and stochastic techniques. All interpolations are based on
the assumption that a point $x_k$ with a known value $z(x_k)$ has a weight on a discrete point $x_0$ in space with an unknown value
$z(x_0)$. The global known points, however, can be reduced to a local neighborhood of $x_0$.

For deterministic interpolation the p-value inverse distance weighting (IDW) (Shepard, 1968) interpolation is used. The IDW
interpolation generally calculates an unknown value $z(x_0)$ at point $x_0$ by weighting the distance of that point to each known
value point $(x_k)$ in space. The underlying formula for IDW is

$$z(x_0) = \frac{\sum_{k=1}^{n} 1/d_k^p \cdot z(x_k)}{\sum_{k=1}^{n} 1/d_k^p},$$   (4)

where $d$ is the Euclidean distance between the point with the known value $x_k$ and the point with the unknown value $x_0$, and
$p$ is an exponent factor to bias the weights non-linearly. The p-value is mostly used for smoothing the results by controlling the
distance-decay effect (Lu and Wong, 2008). IDW is a reliable and widely applied method to interpolate static rock properties
in one to three-dimensional space (Rühaak, 2006).

For stochastic interpolation simple kriging (SK) is used. Kriging in general is a popular technique to interpolate geological
properties in space (Goovaerts, 1997; Rühaak, 2015; Malvić et al., 2019). Through kriging, the value $z(x_0)$ at an unknown
point $x_0$ is calculated by weighting the neighboring known values and building a linear combination of those via the formula

$$z(x_0) = \sum_{k=1}^{n} w_k \cdot z(x_k),$$   (5)

where $w_k$ is the weight of the known point $x_k$ with the value $z(x_k)$. To obtain the simple kriging weights, a set of n equations
has to be solved. This set of equations can be written as

$$\begin{pmatrix} c(x_1 - x_1) & \cdots & c(x_1 - x_n) \\ \vdots & \ddots & \vdots \\ c(x_n - x_1) & \cdots & c(x_n - x_n) \end{pmatrix} \begin{pmatrix} w_1^{SK} \\ \vdots \\ w_n^{SK} \end{pmatrix} = \begin{pmatrix} c(x_1 - x_0) \\ \vdots \\ c(x_n - x_0) \end{pmatrix}$$   (6)

with $c$ as covariance function and $x_n$ as point with known value (Wackernagel, 2003). The quality of kriging interpolation is
dependent on the variogram model, the defined neighborhood, the sampling density and the goodness-of-fit to the experimental
values.





### 2.4 Cross-validation

Cross-validation can be used to assess the quality of a model. During cross-validation, $p$ randomly selected samples are removed from the input data set of size n with $0 < p < n$ and the interpolation is performed without those samples Celisse (2014). The measures of goodness of fit being used in this study include the root-mean-square error (RMSE)

$$RMSE = \sqrt{\frac{1}{n}\sum_{k=1}^{n}\left(\hat{z}(x_k) - z(x_k)\right)^2} \tag{7}$$

and the mean-absolute error (MAE)

$$MAE = \frac{1}{n}\sum_{k=1}^{n}|\hat{z}(x_k) - z(x_k)| \tag{8}$$

with $\hat{z}(x_k)$ as estimated value at point $x_k$. Those parameters allow for the quantitative assessment of an interpolation's quality. They might be prone to bias if the sampling density in the target domain is extremely scarce.

#### 2.4.1 Anisotropy

Anisotropy describes the dependence of a physical property on a direction. Rock properties such as stiffness, permeability or thermal conductivity are anisotropic in most cases. Hence, measurements of those properties might show differing magnitudes in different directions if the medium is polar anisotropic. Anisotropy in geological media is generated by preferred orientation of mineral grains or cracks as well as by the intrinsic anisotropy of single crystals (Thomsen, 1986).

Following, we will exemplary describe the anisotropy of elasticity and we will provide measures for anisotropy quantification under the simplifying assumption of transversal isotropy. The elastic modulus tensor can be expressed as a $4^{th}$-rank tensor

$$\mathbf{C} = \begin{pmatrix} C_{11} & C_{11}-2C_{66} & C_{13} & 0 & 0 & 0 \\ C_{11}-2C_{66} & C_{11} & C_{13} & 0 & 0 & 0 \\ C_{13} & C_{13} & C_{33} & 0 & 0 & 0 \\ 0 & 0 & 0 & C_{44} & 0 & 0 \\ 0 & 0 & 0 & 0 & C_{44} & 0 \\ 0 & 0 & 0 & 0 & 0 & C_{66} \end{pmatrix} \tag{9}$$

where $C_{ij}$ represents an elasticity modulus and the indices are related to the directional p- and s-wave velocity, under the assumption that $z$ is the symmetry axis. The velocities can be calculated by





$$v_p^z = \sqrt{\frac{C_{33}}{\rho}} \qquad (10)$$

$$v_s^z = \sqrt{\frac{C_{66}}{\rho}} \qquad (11)$$

where $v_p$ is the p-wave velocity and $v_s$ is the s-wave velocity parallel to the symmetry axis and $\rho$ is the bulk density (Yang et al., 2020). The anisotropy, here exemplary expressed for the p-wave polar anisotropy, can be quantified with the Thomsen parameters (Thomsen, 1986). For example $\epsilon$ can be expressed as

$$\epsilon = \frac{C_{11} - C_{33}}{2C_{33}}. \qquad (12)$$

If $\epsilon \ll 1$ the material can be classified as weakly anisotropic.

### 2.4.2    Correlation and regression analysis

In order to quantify linear statistical relationship between two independent variables $x$ and $y$, the Pearson linear product-moment correlation coefficient (R) can be used. R is expressed as

$$R = \frac{\sum\limits_{k=1}^{n}(x_k - \overline{x})(y_k - \overline{y})}{\left(\sum\limits_{k=1}^{n} x_k^2 - n \cdot \overline{x}^2\right)\left(\sum\limits_{k=1}^{n} y_k^2 - n \cdot \overline{y}^2\right)}, \qquad (13)$$

with $n$ representing the number of compared point pairs and $\overline{x}$ and $\overline{y}$ standing for the arithmetic mean of $x$ and $y$.

Regression aims at finding a fitting function between samples of two or more random variables. For curvilinear regression, a function of a degree > 1 will be approximated for a discrete set of values. A second-degree polynomial function $f(x)$ for instance would be described as

$$f(x) = b_0 + b_1 x + b_2 x^2 \qquad (14)$$

Thus, we would need to find $n+1$ regression coefficients, where $n$ is the degree of $f(x)$. In general, the regression model yields

$$f(x)_i = b_0 + b_1 x_i + b_2 x_i^2 + \cdots + b_n x_i^n, \qquad (15)$$

with $i = 1, 2, ..., n$. The regression coefficients $b_i$ are obtained through solving a system of linear equations as





$$
\begin{pmatrix} y_1 \\ y_2 \\ \vdots \\ y_n \end{pmatrix} = \begin{pmatrix} 1 & x_1^1 & \cdots & x_1^m \\ 1 & x_2^1 & \cdots & x_2^m \\ \vdots & \vdots & \cdots & \vdots \\ 1 & x_n^1 & \cdots & x_n^m \end{pmatrix} \begin{pmatrix} b_0 \\ b_1 \\ \vdots \\ b_m \end{pmatrix},
\tag{16}
$$

where $x$ and $y$ are the samples. The function approximations as produced in regression analyses are commonly evaluated by the coefficient of determination ($R^2$), which is calculated through

$$
R^2 = 1 - \frac{s_{res}}{s_{tot}} \in [0,1],
\tag{17}
$$

where

$$
s_{res} = \sum_{k=1}^{n} (y_k - f(x)_k)^2
\tag{18}
$$

is the explained sum of squares and

$$
s_{tot} = \sum_{k=1}^{n} (y_i - \overline{y})^2
\tag{19}
$$

is the total sum of squares.

### 2.4.3   Spatial modeling and statistical analyses

The spatial dependence of the discrete values is evaluated through experimental semivariograms. The semivariograms are
generated for the single rock faces, where measurements are available, and for the plug measurements. The empirical semivariogram is fitted by a variogram model, which is then used for the stochastic interpolation. Interpolation analyses are performed as IDW and SK realizations (Fig. 4) that are assessed through cross-validation. The power parameter for IDW is chosen to be three since this constant provides the lowest RMSE among the realizations. The search radii for each prediction is chosen to be 0.2 m in x and y direction and 0.15 m in z direction in OSB1_c to account for the sedimentary structures. For OSB2_c
the search radii are chosen to be isotropic with a length of 0.2 m. To make the methods comparable, we select the maximum number of neighboring points to be 25 representing between 5 and 95% of the measurements.

We decided to waive sequential simulation because major amounts of the cubes' volumes are covered by rock samples. Thus, we do not expect a relevant kriging variance. With this in mind, it is assumed to capture most of the total variance by the measurements themselves. The interpolation results that provide the lowest cross-validation error are used for statistical
analyses in order to derive correlations and regression functions between the scalar fields. Eventually, significant correlations


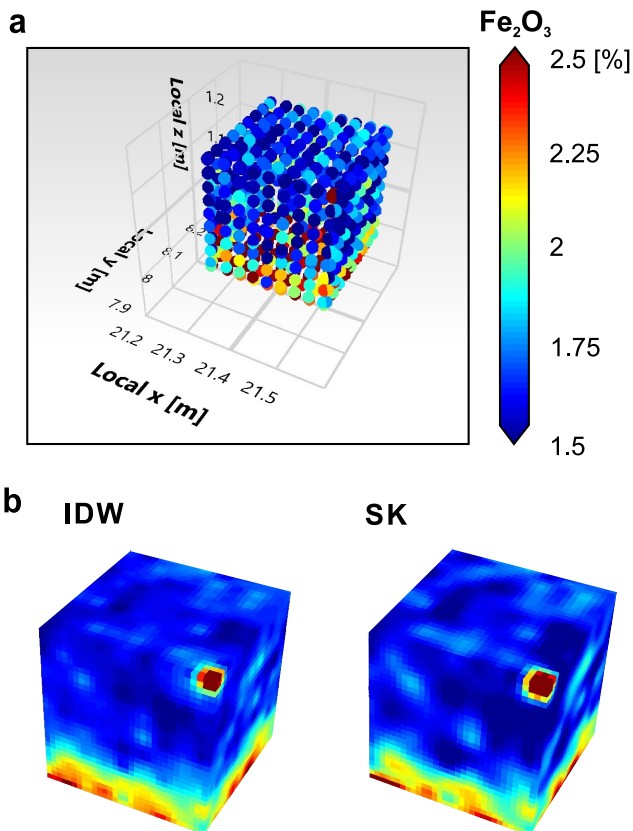

**Figure 4.** (a) $Fe_2O_3$ measurement locations on the cube faces of OSB1_c and on the rock samples extracted from the cube. The diameter of one point is $1.2\,\mathrm{cm}$, which corresponds to the beam diameter of the pXRF measurement device. (b) Visual representation of the inverse distance weighting (IDW) and simple kriging (SK) realization of the 3-D scalar field of $Fe_2O_3$ using the the discrete points displayed in (a) as known data points.

are compared with the non-interpolated data sets. Both the spatial modeling and the statistical analyses are performed with the open-source software GeoReVi (Linsel, 2019). This software tool provides functionality for multidimensional subsurface characterization using the concept of knowledge discovery in databases, which is helpful when handling huge data sets as produced in this study.

## 3 Results

### 3.1 Sedimentological characteristics

The sandstones belong to a clinothem strata deposited in a fluvial-dominated lacustrine delta. More specific, the architectural element represents a distributary mouth bar, formed by rapid sandstone deposition in sheet-like bodies like exemplary described





in (Fongngern et al., 2018). The base of those bodies is typically erosive, which is why muddy rip-up clasts commonly occur

above the base. Also, the beds, which deposited after the intraclast-rich basal beds, typically show trough or ripple-cross stratification with set heights of 5–15 cm. Vertical orientation of rip-up clasts can be observed in matrix-rich debrites or turbidites deposited under high-energy turbulent hyperpycnal to homopycnal flow conditions (Li et al., 2017). Those are unconformably overlying lacustrine, laminated mud strata from the prodelta environment. Accordingly, Bouma A-C layers (Bouma, 1962) with a prograding trend can be identified in the outcrop. The beds, from which the cubes were taken, correspond to Bouma

A. With ongoing sedimentation, the depositional energy in a Bouma A sequence typically decreases, which leads to massive sandstones. OSB1_c was taken from a basal bed of the Bouma A sequence, characterized by a high amount of intraclasts, normal grading and sub-horizontal layering, whereas OSB2_c was taken from the topmost bed, characterized by a homogeneous structure.

  The average grain size in both cubes ranges from fine to very coarse sand (200–1400 μm). While the grain size distribution

in OSB2_c does not show a significant variability – mainly characterized by medium to coarse sand – a normal grading is observable in OSB1_c. Here, the grain size gradually transitions from very coarse sand at the base to medium sand at the top. Likewise, sorting transitions from poor to moderate sorting. In OSB2_c the sorting is moderate continuously. The components provide a low to medium sphericity while the grain shapes vary between sub-angular to sub-rounded. Locally, pelitic rip-up clasts occur with diameters of up to four centimeters. The rip-up clasts show a very low textural maturity and are oriented

sub-vertically.

  The original rigid detrital components consist of 50–60% quartz, 20–30% strongly altered feldspar as well as micas and lithic fragments. Mica grains are often bent between more rigid grains. The rock matrix accounts for approximately 10–20% and is built up by detrital grains – coated by iron oxides –, ductile, autochthonous grains and fine-grained quartz. According to the geochemical analyses, the rocks can be classified as lithic arenites to arkoses or wackes (Fig. 5) respectively, if the matrix

content exceeds 15%, applying the classification of (Herron, 1988).

  Thin section analysis (Fig. 6a) reveals that most of the pore space is produced secondary due to grain dissolution. The secondary pores are undeformed indicating that grain dissolution took place during structural inversion – probably during telogenesis according to the concept of Worden and Burley (2003). Most of the inter-granular volume was destroyed during mechanical compaction. ESEM analysis (Fig. 6b) confirms the presence of quartz accompanied by co-precipitated calcite,

opaque phases – mainly iron oxides – and authigenic clay minerals including kaolinite and illite in the cement fraction. Thus, chemical compaction had taken place by iron oxide, quartz and clay mineral precipitation during diagenesis. Here, the earliest cement phase is represented by the opaque phases comprising a high amount of iron oxides. Following, kaolinite is formed, mainly in the secondary pore space, overgrown by illite. Often, the early cement is overgrown syntaxially by quartz. The source of $SiO_2$ might be internal and related to feldspar dissolution.





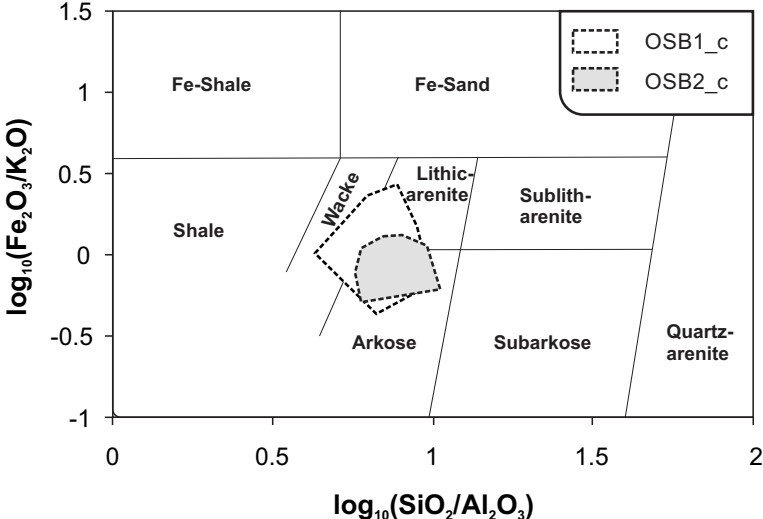

**Figure 5.** Petrographic classification after Herron (1988) based on the ratio of $SiO_2$ and $Al_2O_3$ and $Fe_2O_3$ and $K_2O$. The polygons show the convex hull for the measurements derived from the cubes' faces.

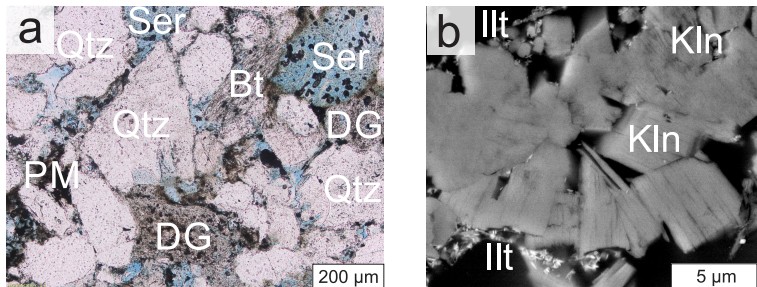

**Figure 6.** (a) Representative thin section taken from rock cube OSB2_c. The sandstone consists mainly of quartz (Qtz), altered feldspars with residual mineral aggregations (sericite, Ser), altered biotite (Bt) and ductile grains (DG). Feldspar dissolution lead to a high grade of secondary porosity (Molenaar et al., 2015) while major parts of the intergranular pore space are filled with primary and pseudomatrix (PM), which is rich in iron oxides. (b) Environmental scanning electrone microscope (ESEM) image of the authigenic clay minerals (mainly kaolinite (Kln) and illite (Ilt)) built in the pore space. Mineral abbreviations were taken from Whitney (2010).

## 3.2 Exploratory data analysis

In order to provide full comparability, the following section will provide an overview over the measurements derived from the rock cylinder measurements. For each property, 79 rock samples from OSB1_c and 29 from OSB2_c were investigated. An overview over the properties' ranges is provided in the Box-Whisker charts displayed in Figure 7.

The local variability of OSB1_c is significantly higher than that of OSB2_c. Intrinsic permeability of OSB1_c provides a coefficient of variation of 0.3 and a Dykstra-Parson coefficient of 0.4 while measurements from OSB2_c show values of 0.2 for





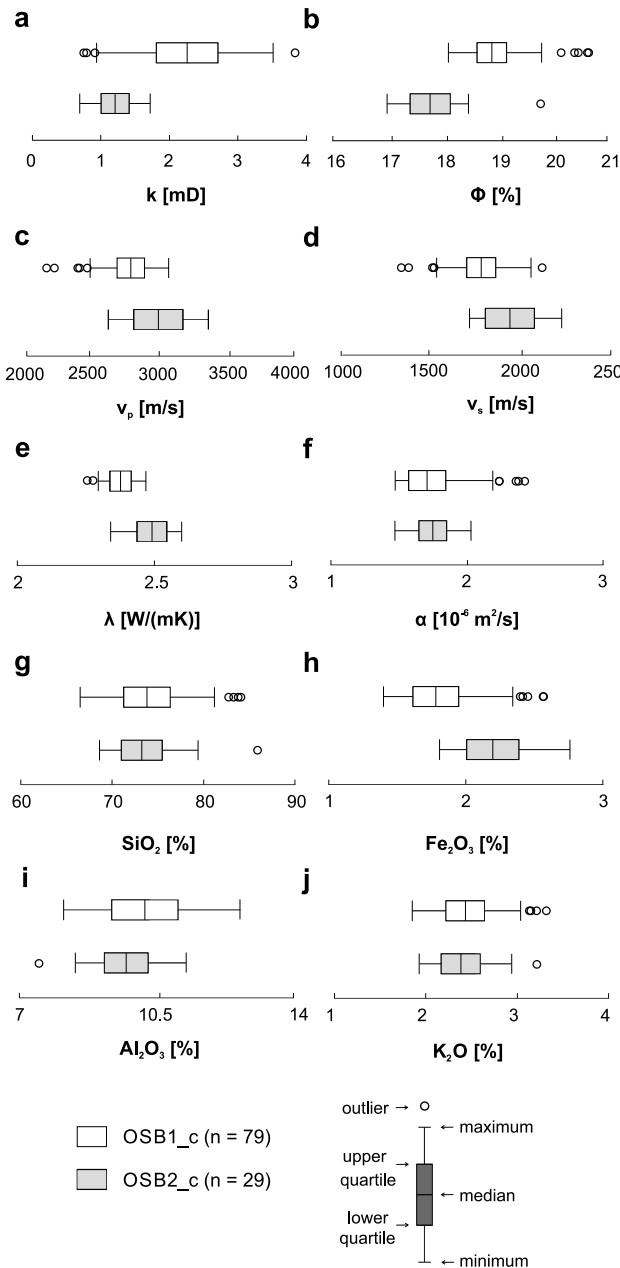

**Figure 7.** Box-Whisker charts showing the empirical distribution of the rock properties measured on the rock cylinders taken from the rock cubes. Outliers were detected according to Tukey's method (Tukey, 1977) where a value is tested to be in the 1.5-times inner-quartal-range of the arithmetic mean. (a) Intrinsic permeability k (b) Effective porosity $\phi$ (c) p-wave velocity $v_p$ (d) s-wave velocity $v_s$ (e) Thermal conductivity $\lambda$ (f) Thermal diffusivity $\alpha$ and the mass fraction of (g) Silicon oxide $SiO_2$ (h) Iron oxide $Fe_2O_3$ (i) Aluminum oxide $Al_2O_3$ and (j) Potassium oxide $K_2O$.





the coefficient of variation and 0.18 for the Dykstra-Parson coefficient respectively. According to the classification provided by Corbett and Jensen (1992), the intrinsic permeability of both rock cubes can be classified as being very homogeneous.

The range of values in OSB1_c for each property is greater than the range of those in OSB2_c. OSB1_c provides lower values in p- and s-wave velocity, thermal conductivity and mass fraction of $Fe_2O_3$ compared to OSB2_c. Intrinsic permeability
and porosity in turn are greater. The mass fraction of silicon oxide and thermal diffusivity provide similar statistical parameters in both cubes, however, the ranges are marginally larger in OSB1_c. The measurements of the elastic rock properties revealed a weak anisotropy of the p-wave attenuation especially in rock cube OSB2_c. The Thomsen parameter $\epsilon$ is 0.047 for OSB1_c and 0.096 for OSB2_c. It should be noted that OSB1_c provides visible bedding structures in contrast to OSB2_c, hence, the observed degree of anisotropy is not connectable to the bedding features in this case.

Statistically significant linear correlations (Fig. 8), in the sense of passing a two-tailed significance test at the 0.05 level, were found between porosity and permeability, permeability and $Fe_2O_3$, $v_p$ and $v_s$, $v_p$ and $SiO_2$, $v_p$ and $Al_2O_3$, $v_p$ and $K_2O$, $Fe_2O_3$ and $SiO_2$ as well as $K_2O$ and $Al_2O_3$. The strongest positive linear correlation can be observed between $v_p$ and $v_s$ (R = 0.88), $K_2O$ and $Al_2O_3$ (R = 0.70) and porosity and permeability (R = 0.31). The strongest negative correlation can be observed between permeability and $Fe_2O_3$ (R = -0.56). Properties not being mentioned do not provide significant statistical correlations
to others.

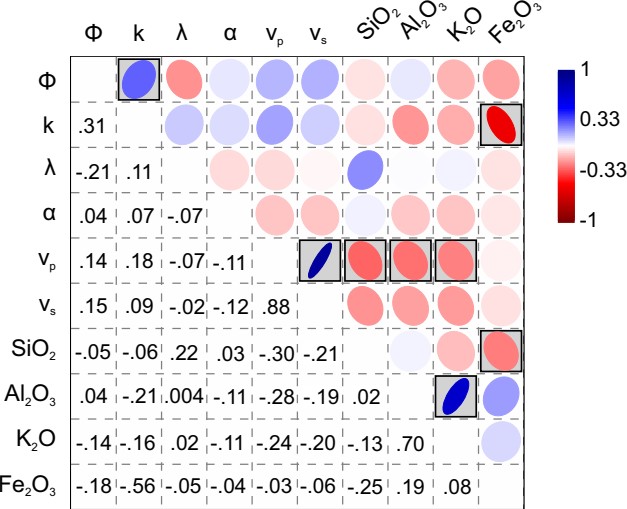

**Figure 8.** Matrix visualization of the Pearson correlation coefficient derived from the plug measurements. Statistically significant correlations with a p-value $\leq$ 0.05 are highlighted by gray boxes. The diameter of the ellipses' conjugate axes is dependent on the correlation coefficient. The smaller the length of the axis, the stronger is the correlation. The matrix is diagonal meaning that the Pearson correlation coefficient as numerical expression is located at the diagonal position relative to each ellipsis. $\Phi$ = effective porosity; k = permeability; $\lambda$ = thermal conductivity; $\alpha$ = thermal diffusivity; $v_p$ = p-wave velocity; $v_s$ = s-wave velocity.





### 3.3 Sub-meter scale spatial correlation

The spatial dependence of the discrete measurements is estimated using experimental semivariograms. Therefore, the geochemical representatives $SiO_2$ (Fig. 2a) and $Fe_2O_3$ (Fig. 2b) that were measured on each of the rock faces of OSB1_c are therefore exemplary analyzed. The experimental semivariograms greatly vary from face to face in OSB1_c. The nugget effect
for each experimental variogram is very low. The range of each semivariogram varies between 0.05 and 0.3 m. In the experimental semivariograms of $SiO_2$, two types of patterns can be identified. The faces XY Base, XZ Back and YZ Front, which are displayed in Figure 9, all show ranges of approximately 0.08 m and a sill between 8 and $10\%^2$ until the semivariance exponentially increases when exceeding a lag distance of 0.2 m. The semivariance on the other faces runs similarly with ranges of 0.2 m and a sill of $4.7\%^2$. The semivariogram for $Fe_2O_3$ shows some similarities. Here, the faces XY Base, YZ Front and
XZ Front show very low ranges between 0.05 and 0.15 m and sill between 0.1 and $0.15\%^2$ again with an exponential increase when exceeding a lag distance of 0.2 to 0.25 m. In contrast, the semivariance of YZ Back has the highest sill with $0.21\%^2$ and a range of 0.15 m, however, semivariance drops after exceeding a lag distance of 0.2 m. XZ Back provides the highest degree of similarity with a range of 0.3 m and a sill of $0.09\%^2$ using a spherical approximation. Both geochemical properties show a zonal anisotropy where the sill shows different magnitudes along different directions (Wackernagel, 2003; Allard et al., 2016).

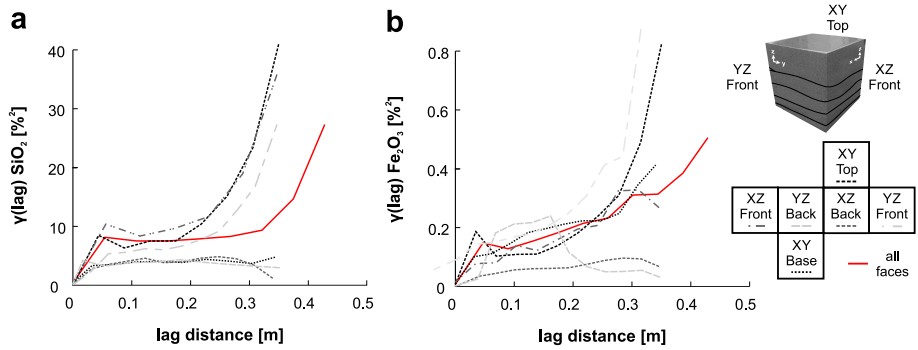

**Figure 9.** Empirical semivariograms of the mass fraction of $SiO_2$ (a) and $Fe_2O_3$ (b) in rock cube OSB1_c grouped by the investigated rock face.

### 3.4 Spatial pattern analysis

The spatial distributions of the rock properties are interpolated with Shepard's inverse distance weighting (IDW) and simple kriging (SK). Both realizations of a single scalar field provide comparable patterns. The interpolation errors are also located in similar ranges, however, IDW seems to be more sensitive to outliers resulting in much higher interpolation errors with regard to properties like p-wave velocity or mass fraction of $SiO_2$ (Table 1). IDW tends to underestimate the maximum and minimum
values in the scalar fields. Thus, petrophyscial and geochemical contrasts are more distinctly reproduced in the stochastic approach. Also, the IDW realization shows the bull's eye effect, which is a typical artifact of IDW interpolations (Shepard, 1968). Accordingly, the simple kriging realizations are used for further analyses.





The rock properties exhibit a multitude of spatial patterns. Here, discrete, layered and homogeneous patterns, both connected and disconnected to primary sedimentary structures, could be observed in the interpolations.

**Table 1.** RMSE and MAE for the interpolation results of IDW and SK for OSB1_c. k = permeability; $\Phi$ = effective porosity; $\lambda$ = thermal conductivity; $\alpha$ = thermal diffusivity; $v_p$ = p-wave velocity.

|  | RMSE IDW | RMSE SK | MAE IDW | MAE SK |
|---|---|---|---|---|
| k | .19 | .17 | .15 | .14 |
| $\Phi$ | .54 | .59 | .4 | .42 |
| $\lambda$ | .23 | .22 | .18 | .16 |
| $\alpha$ | .14 | .17 | .1 | .1 |
| $v_p$ | 64.19 | 60.95 | 52.21 | 44.74 |
| $SiO_2$ | 4.07 | 3.25 | 3.05 | 2.09 |
| $Al_2O_3$ | .8 | .83 | .66 | .66 |
| $K_2O$ | .25 | .26 | .19 | .2 |
| $Fe_2O_3$ | .93 | .32 | .86 | .21 |

### 315 3.4.1 Patterns connected to sedimentary structures

A bedding-connected pattern is exhibited in the intrinsic permeability and $Fe_2O_3$ interpolation results of OSB1_c. The mass fraction of $Fe_2O_3$ varies between 1.25 and 5% in OSB1_c. In the histogram displayed in Figure 11 outliers were removed according to Tukey's outlier detection method (Tukey, 1977). The local histogram of OSB1_c's intrinsic permeability shows a bimodal distribution ranging from 0.7 to 3.9 mD. The application of Tukey's method revealed no statistical outliers in this
scalar field.

The bedding structures in OSB1_c are well reflected by the spatial pattern of the interpolated intrinsic permeability gradually increasing from low values between 0.7 and 2 mD in the lower beds to higher values between 2 and 4 mD in the upper beds (Fig. 10).

The spatial distribution of the mass fraction of $Fe_2O_3$ in OSB1_c provides a reciprocal trend compared to the permeability.
Here, the lowermost bed shows a significantly higher content compared to the upper beds. Both scalar fields show zonal anisotropy. The $Fe_2O_3$ content is an indicator for the detrital matrix, pseudomatrix and cement content what in turn would explain the reciprocal relationship with the permeability measurements. In siliciclastic systems, iron can be contained in clay minerals (up to 30wt% (Brigatti et al., 2006)), mafic components or in iron-rich oxides, hydroxides or carbonates. Local excesses in the $Fe_2O_3$ content exist in the spatial distribution. Those can be explained by clay-rich intraclasts observed on the
rock faces. When comparing the pattern to Figure 2 at both XZ-oriented cube faces, rip-up clasts can be observed where high $Fe_2O_3$ mass fractions occur. Those areas provide the maximum values of the $Fe_2O_3$ distribution.




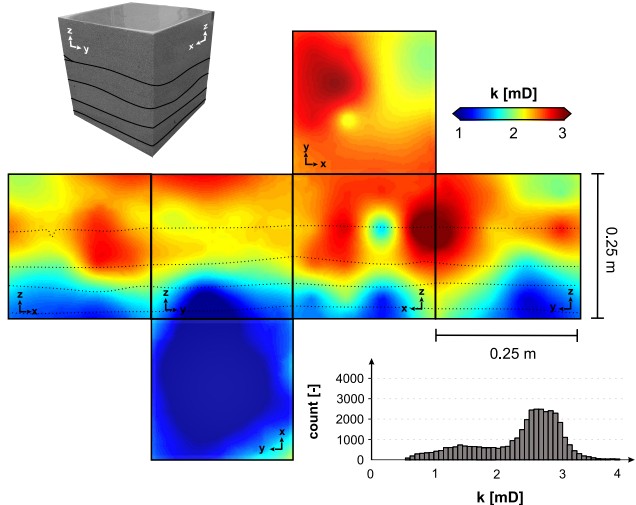

**Figure 10.** Spatial distribution of the intrinsic permeability modeled with a simple kriging interpolation. The histogram shows a bimodality of the distribution split up into the basal beds and uppermost beds.

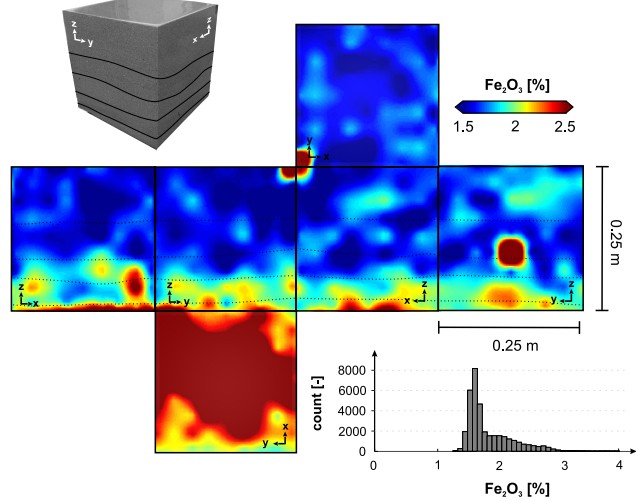

**Figure 11.** Spatial distribution of the mass fraction of $Fe_2O_3$ modeled with a simple kriging interpolation. As in the intrinsic permeability interpolation, a bimodality can be observed in the empirical histogram.

### 3.4.2 Patterns decoupled from sedimentary structures

Other scalar fields are decoupled from depositional bounding surfaces. For instance, the geochemical mass fractions of $K_2O$ (Fig. 12) and $Al_2O_3$ (Fig. 13) provide a significant positive correlation unconnected to visible structural boundaries. Typically,
those geochemical properties are indicative for the presence of orthoclase feldspar ($KAlSi_3O_8$) and/or illite ($KAl_3Si_3O_{10}(OH)_2$)



in siliciclastic environments. The mass ratio of both components is roughly 1:3 to 1:4, which is in accordance to the illite frac-tion that was observed in thin section and ESEM analyses. Only minor amounts of orthoclase feldspar could be found in the thin sections. Thus, we assume that the correlation of $K_2O$ and $Al_2O_3$ can traced back to the illite phases.

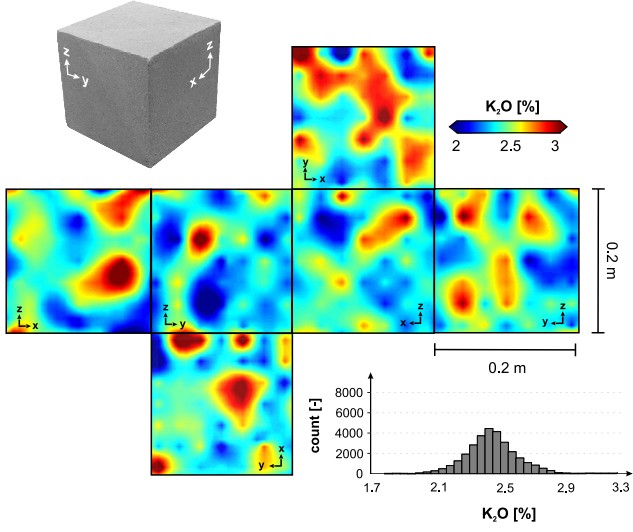

**Figure 12.** Spatial distribution of the mass fraction of $K_2O$ modeled with a simple kriging interpolation. The pattern is decoupled from primary sedimentary structures and shows a network-like structure.

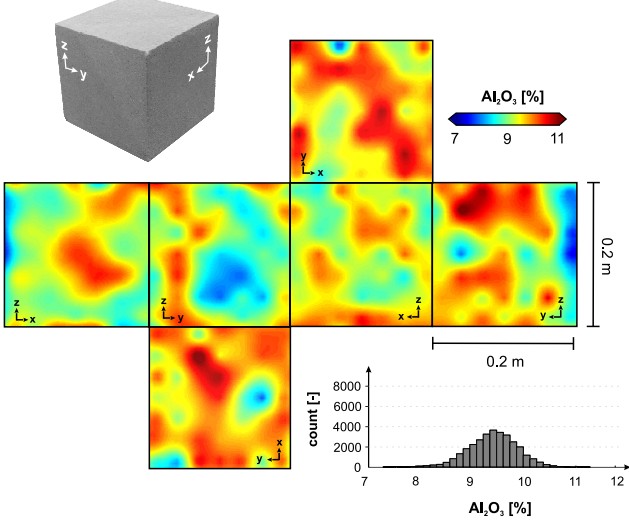

**Figure 13.** Spatial distribution of the mass fraction of $Al_2O_3$ modeled with a simple kriging interpolation. The pattern is decoupled from primary sedimentary structures and shows a network-like structure.





Higher fractions of $Al_2O_3$ are supposedly due to higher kaolinite ($Al_2Si_2O_5(OH)_4$) fractions in the clay mineral assemblages.
The patterns are diffuse showing autocorrelated areas of slightly enriched and depleted mass fractions. Enriched areas seem to be connected, building network-like patterns, while depleted areas are more isolated.

## 4 Discussion

The overall aim of this study was to quantify the three-dimensional inter-dependencies of thermophysical, hydraulic, elastic and geochemical scalar fields in sandstone media at the lithofacies scale and to identify the controlling factors for the property dis-
tributions. With a high-resolution study at the lithofacies scale, statistical and spatial inter-relationships between characteristic physicochemical fields could be discovered and traced back to depositional and diagenetical processes.

### 4.1 Petrophysical and geochemical characteristics

Recent multi-scale modeling approaches without the usage of local constraints show that the prediction of permeability and porosity in siliciclastic systems is still challenging (Nordahl et al., 2014). Geological sampling almost never includes the
entire domain that is investigated. With sampling densities of 25.4% and 18.2%, we reached a very high degree of coverage. Studies such as Hurst and Rosvoll (1991) showed that a very high sampling density is necessary to cover the entire variance of permeability at the lithofacies scale. The interpolations performed in this study reproduce the global histogram properly and also outliers are accounted for. This, in fact, implies that the sampling density was selected adequately in order to capture the total variability present in the physical and geochemical scalar fields. This condition is typically only fulfilled in sequential
simulations (Robertson et al., 2006) rather than in conventional interpolations.

Although statistically significant correlations may imply a natural relationship between physicochemical properties, this relationship could also be based on random processes requiring causality to be verified. Weak correlations were found between the effective porosity and the intrinsic permeability, which are positively correlated usually (Pape et al., 1999). This relationship can be traced back to the Kozeny–Carman equation that connects the permeability with the effective pore throat radius $r_{eff}^2$
and a formation factor $F$ like

$$k = r_{eff}^2/(8 \cdot F). \tag{20}$$

The formation factor is defined as the ratio of tortuosity and porosity showing that porosity and permeability provide a positive formal relationship empirically. A high amount of secondary pores, produced by feldspar dissolution, did not significantly contribute to the permeability in the investigated sandstones since those pores are often hydraulically isolated. Consequently,
secondary porosity did not necessarily lead to increasing radii of the effective pore throats rather than increasing tortuosity. Also, recrystallized quartz cement – blocking a large amount of the pore throats – must be taken into account. Both effects, in turn, resulted in a degraded permeability. Additionally to the geometrical aspects previously mentioned, the alteration products in form of clay minerals occupy the pore space, which leads to larger adhesive effects that hinder the ability to transport fluids





as well. This observation is in good agreement with observations made by Molenaar et al. (2015) in Rotliegend rocks from the Donnersberg formation.

The linear correlation analysis revealed a significant negative relationship between hydraulic and geochemical properties that fits to a polynomial regression (Fig. 14). It should be considered that the geochemical measurements cover a very different measurement area – represented by a spot of 1.2 cm diameter and around 0.5 cm penetration depth compared to the hydraulic measurements performed on an entire rock cylinder of 40 mm height and diameter. Additionally, instead of using highly-precise stationary X-ray fluorescence devices for measurements, a portable, faster device was used to efficiently derive spatial trends in the objects of investigation. This technique weakens the implications for absolute values, however, the trends observed in the measurements from the portable device are in good agreement with trends observed by stationary devices. Also, the observed geochemical characteristics are in well accordance with geochemical properties of quartz-rich sandstone varieties that were investigated in Bhatia (1983) or Baiyegunhi et al. (2017).

Geochemical analyses, in contrast to petrographic ones, limit the interpretations of geological processes because mineral phases can only be assumed and not determined for certain. A high mass fraction of $Fe_2O_3$ may imply that the rock is rich in iron-bearing minerals like clay minerals, hematite, magnetite, goethite, lepidocrite or ferrihydrite (Costabel et al., 2018), however, a precise classification of the mineral phase is not possible. Iron oxides are more common in secondary precipitates that usually form during eo- and mesodiagenesis (Pettijohn et al., 1987). The degrading impact of iron-oxide-rich coatings on permeability and porosity in unconsolidated sand and gravel has been shown in studies like Costabel et al. (2018). The amount of detrital iron-rich phases like hematite present in the rock matrix is typically less (Walker et al., 1981; Turner et al., 1995) compared to the secondary amount. In our case, however, thin section and ESEM analyses revealed that a high degree of intergranular matrix is still preserved, especially at the base of OSB2_c where high amounts of mud and mud intraclasts are incorporated from basal erosion. The small grain size of the matrix offers a great surface area for iron-oxide-rich precipitates, which might have enforced degradation of porosity and permeability additionally. Primary matrix typically plugs the pore throats of porous media, which reduces the ability to conduct fluids compared to matrix-free analogies. However, due to progressive compaction we cannot quantify for certain how large the amount of the primary matrix is compared to the pseudomatrix produced by plastic compaction of ductile, clay-rich grains and by feldspar dissolution.

A significant correlation between $K_2O$ and $Al_2O_3$ could be detected. The spatial distribution resembles a network-like structure that might be either a product of diffusive mass transport during meso- or telodiagenesis or might reflect the distribution of feldspar grains and its residues in the sandstone. During feldspar alteration, $SiO_2$ gets dissolved and K remains in the alteration products, which could be an implication for the meso-scale network-like structure, in which pore fluids could have had migrated. This relationship is underlined by a negative, yet non-significant correlation of $K_2O$ with $SiO_2$.

## 5  Conclusions

Significant, non-intuitive relationships between the physical and geochemical scalar fields at the lithofacies scale have been revealed with a deductive approach of spatial field modeling and statistical data analysis.





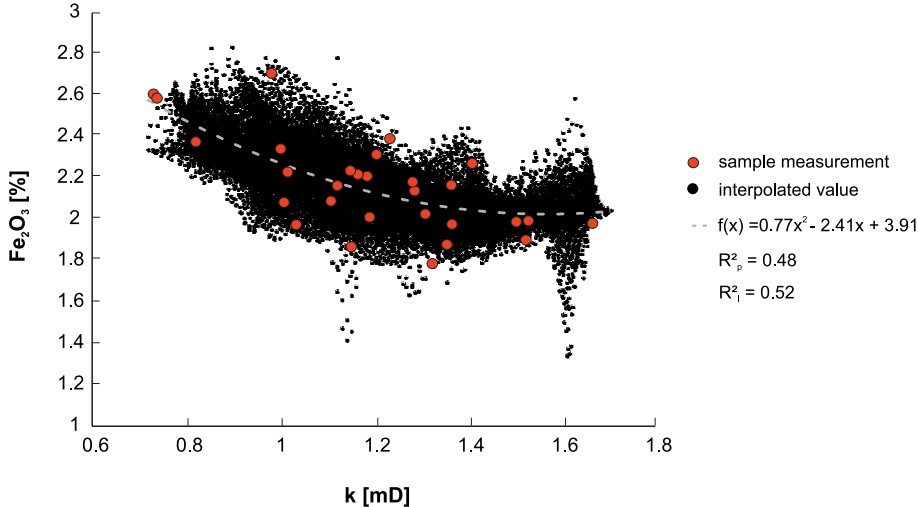

**Figure 14.** Regression analysis of the relationship between intrinsic permeability and mass fraction of $Fe_2O_3$ in the interpolated scalar fields of the rock cube OSB2_c. $R_p^2$ is the coefficient of determination for the plug measurements and $R_i^2$ is the the coefficient of determination for the interpolated values.

1. As specific properties like the mass fraction of $Fe_2O_3$ preserve sedimentological textures well in their spatial distribution, other properties seem to be completely decoupled from depositional bounding surfaces. These scalar fields probably reflect processes that might have had taken place during diagenetical overprint of the rocks as a result of burial and exhumation. These processes produce diffuse patterns, like discussed with regard to the correlation of $K_2O$ and $Al_2O_3$.

2. This study demonstrates that the observation of bedding structures does not necessarily indicate a stronger polar anisotropy compared to macroscopically unstructured lithologies. Here, the microscopic characteristics like the amount of secondary porosity might play a more important role in the attenuation of physical waves than the bounding surfaces.

3. It could be shown that hydraulic properties are dependent on the intergranular matrix and cement amount, which are in turn controlled by depositional processes and eogenetical precipitates. Those findings are not new (see Wilson and Pittman (1977) or Nordahl et al. (2014)), however, have not been evaluated in lithofacies-scale 3-D environments yet. We assume that primary matrix and ductile grain content has the most detrimental effect on rock permeability. Ductile grains were mechanically deformed during compaction leading to plugged pore throats. Feldspar dissolution has a highly productive effect on porosity but not on permeability.

4. We demonstrate that the strength of statistical correlation can be preserved in spatial interpolations as long as the sampling density is sufficient. If the sampling density is too low, a statistical correlation might be feigned inadvertently.



In the future work, we will examine in detail how the patterns observed in the major elements' mass fractions are connected to diagenetical processes. Therefore, it is planned to perform XRD analyses. In addition, we are planning to model the pore network within the rock cubes as a major controlling factor for diffusive mass transfer with the help of μCT recordings.

420   *Code and data availability.*   GeoReVi is an open-source software for Windows systems available under https://github.com/ApirsAL/GeoReVi. The executables are available in the repository under https://github.com/ApirsAL/GeoReVi/blob/master/binaries/. The measurements are available under https://www.doi.org/10.6084/m9.figshare.11791407.v2

*Sample availability.*   The investigated rock samples are available at the Institute of Applied Geosciences Darmstadt and can be requested under linsel@geo.tu-darmstadt.de. Also, the samples are registered in the System for Earth Sample Registration (SESAR, www.geosamples.org).

425   *Author contributions.*   AL conceptualized and prepared the manuscript. AL and SW conducted the laboratory and field measurements. JH contributed to the conceptualization of the study. MH was the overall supervisor of the study.

*Competing interests.*   The authors declare that they have no conflict of interest.

*Acknowledgements.*   The authors are grateful for the permission to work in the sandstone quarry of the company Konrad Müller GmbH in Obersulzbach, Germany. AL received funding from the Friedrich-Ebert-Stiftung, Germany.





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
