# Peer review of "High-Resolution Analysis of the Physicochemical Characteristics of Sandstone Media at the Lithofacies Scale"

_Solid Earth, 2020_

## Referee Comment (RC1) · Mattia Pizzati (Referee) · 28 May 2020

Review of manuscript **se-2020-13** submitted to Solid Earth, Linsel et al. "**High-resolution analysis of physicochemical characteristics of sandstone media at the lithofacies scale**" by Mattia Pizzati.

**General remarks**

Dear Authors and Editor, below you can find the review of the submitted manuscript. Revisions are made by describing the issues found line by line and also on the text file of the manuscript, in which critical points were highlighted in green color. Comments on figures, figure captions and tables are presented at the end of this file as well.

This manuscript is focused on the analysis of physical, chemical petrophysical and diagenetic properties characterizing a fluvial-lacustrine sandstone mouth bar at the sedimentological facies scale. Applied methods include both field and laboratory measurements, statistical validation and cross-checking.

Laboratory analyses and statistical tests are thoroughly described and explained, and even if the section describing methods may appear quite long compared to the rest of the manuscript, I feel it is necessary to the full comprehension of the results and discussion. This is particularly true for readers who may not have a strong statistical background and without this support they could be discouraged in going through the text.

The quality of written English is very high, sentences are clear and reading is fluent. I found some minor mistakes and added a few suggestions to further improve the structure of sentences.

Figures are schematic, clear and of good quality. I inserted a few suggestions aimed to improve their visibility and minor corrections to be done mainly on legends and captions.

Data presented fully support the thesis the authors wanted to discuss. This manuscript represents a good example of how important is the detailed study of petrophysical, diagenetic and physical properties of sandstone at the scale of the sedimentary facies. Maybe it would have been nice to have an additional paragraph inside the discussion in which the authors could have described the implications for water-oil reservoir exploitation and maybe also the up-scaling of the properties they discuss. However, this could be much beyond the original aim of the present work, but it may represent a cue for future studies.

Another point that could be of real interest is to test the differences or similarities between the properties of undeformed sandstone and fault rocks, since in the described outcrop at least two faults with few meter of displacement are present. Again consider this as a hint for next research topics.

I think this paper is worth to be accepted on Solid Earth after the completion of technical-minor revisions.

I really hope this review will be helpful to the improvement of the final version of your manuscript. It is a nice research topic and deserves to be developed in the best way.

Do not hesitate to contact me in case any questions or doubts arise from these comments: mattia.pizzati@studenti.unipr.it

**Detailed comments line by line**

Line 14: Please add a hyphen between "*intergranular*" to "*inter-granular*".

Line 19: Perhaps this is just a tiny detail, but also in textbooks I always read "P and S-waves" with capital letters. In the entire text of the manuscript it is reported with lower case. Consider if this suggestion suits you. To ease the identification, I underlined "p and s-waves" throughout all the text.

Line 20: Here maybe "*strictly*" may sound better than "*inevitably*".

Line 22: Consider substituting "*In fact*" with "*Following this,*".

Line 50: Please add a hyphen in "*interrelationships*" to "*inter-relationships*".

Line 54: Please change "*are taken*" with "*have been sampled*".

Line 56: Perhaps the term "*series*" sounds a bit too generic. Is it possible to adopt the word "*orogeny*" or "*tectonic event*"? I have limited knowledge of regional geology of Germany, so evaluate if this term need to be corrected or not.

Line 59: Check if you want to keep the lower case version "*p- and s-waves*" or the capitalized one "*P- and S-waves*".

Line 72: When you are referring to specific hierarchical stratigraphic nomenclature (formation, group and so on) you should adopt the capital letters. Following this, please correct " *Disibodenberg formation* " with "*Disibodenberg Formation*".

Line 73: You provided the present-day depth of the Disibodenberg Formation in the Upper Rhine Graben. Is it possible to constrain, even roughly, the maximum burial depth experienced by the studied sandstone in your field site? It could be very interesting since further below you describe the effects of mechanical and chemical compaction affecting the sandstone.

Line 76: To avoid the repetition of the word "*formation*" you can substitute the second highlighted word simply with "*it*".

Line 77: Here maybe "*were sampled*" or "*were extracted*" sounds better than "*were taken*".

Line 81: Please change the sequence of these words from "*The cubes both were...*" to "*The cubes were both...*".

Line 84: With the term "*granularity*" are you describing the angularity of grains composing the mouth-bar sandstone strata? If so I believe that "*angularity*" provides a clearer and more straightforward idea of what you are describing.

Lines 86-87: Since you are describing a methodology with only one bibliography reference maybe you should turn to the singular form "*This approach, however, does often...*".

Line 88: Instead of "*we measured the faces of the cubes...*" if you prefer adopt the form "*we performed analyses on faces of the cubes...*".

Line 96: Can you define the size of the elementary cell used in 3D models?

Line 97: Perhaps the title of the paragraphs could be changed to "*In situ measurements*" or "*Field measurements*".

Line 102: Check if you want to keep the lower case version "*p- and s-waves*" or the capitalized one "*P- and S-waves*".

Line 105: Check if you want to keep the lower case version "*p- and s-waves*" or the capitalized one "*P- and S-waves*".

Line 111: Since in the same sentence you used the past simple, I would recommend to keep the same style. If you agree change "*do*" with "*did*".

Line 117: Check if you want to keep the lower case version "*p- and s-waves*" or the capitalized one "*P- and S-waves*".

Line 119: Here maybe erase the second "*travel*" since it could be a repetition.

Line 119: Check if you want to keep the lower case version "*p- and s-waves*" or the capitalized one "*P- and S-waves*".

Line 129: Perhaps "*insights on*" sounds better than "*insight into*".

Line 130: I would rework the sentence as indicated: "*can significantly impact the petrophysical properties*".

Line 151: Check if here you want to use "*neighborhood*" or "*neighbor*".

Line 186: Check if you want to keep the lower case version "*p- and s-waves*" or the capitalized one "*P- and S-waves*".

Lines 191-192: Check if you want to keep the lower case version "*p- and s-waves*" or the capitalized one "*P- and S-waves*".

Line 244: Consider if "*sampled*" suits better than "*taken*".

Line 247: In the previous sentence you state that the basal part of the bed can be classified as Bauma A subdivision, which ideally implies massive medium-coarse sandstone. However, in this line it is reported the presence of "*sub-horizontal layering*", that may contrast with what is described above (massive should mean structureless). Can you please clarify this point? I fully understand that the Bauma subdivision is something purely ideal, and differences from the model may occur. Are these "*sub-horizontal layers*" similar to crude laminations, which typically occur in the upper part of the Bauma A subdivision?

Line 247: By "*homogeneous*" here you mean massive, without any structure or are you referring to the grain size. Sorry for being blunt here, but personally "*homogeneous*" fells a bit to generic.

Line 251: Here maybe instead of "*transitions*" you can use "*decreases*".

Line 252: See if this reworking suits you: "*Likewise, sorting increases from poor to moderate*".

Line 252: Instead of "*continuously*" perhaps use "*throughout the entire sample volume*".

Line 255: To be more explicit please add "... *sub-vertically with respect to bedding*".

Line 256-257: Could you please separate the percentage of feldspar and the one pertaining to the lithic fragments (mica, rock fragments)? Just to have the percentage relative to quartz, feldspar and rock fragments.

Line 258: Please define if possible the nature of "*ductile, autochthonous grains*".

Line 260: Here I believe you can use the in-text citation reference "*Herron (1988)*" instead of the form in parentheses.

Line 261: Here I would erase "*produced*".

Line 263: You can add "*primary*" before "*inter-granular*" since here you are referring to the original porosity of the sandstone prior to compaction.

Line 272: You can use the word "*analyses*" to avoid repetition of "*measurements*".

Line 279: Check if you want to keep the lower case version "*p- and s-waves*" or the capitalized one "*P- and S-waves*".

Line 282: Check if you want to keep the lower case version "*p- and s-waves*" or the capitalized one "*P- and S-waves*".

Line 309: Check if you want to keep the lower case version "*p- and s-waves*" or the capitalized one "*P- and S-waves*".

Line 310: Please correct the misspelled word "*petrophyscial*" with "*petrophysical*".

Line 326: Here I would erase "*The*" to make the sentence lighter.

Line 326: Maybe "*that*" is more correct than "*what*".

Line 368: Here you should use the plural form "*lead*" instead of the third person "*leads*", since the subject of the sentence is plural "*alteration products*".

Line 388: Please add a hyphen between "*intergranular*" to "*inter-granular*".

Line 390: See if this sentence reworking sounds: "*...which might have further enforced degradation of porosity and permeability*".

Line 391: Maybe you can change "*analogies*" simply with "*ones*".

Line 409: Please add a hyphen between "*intergranular*" to "*inter-granular*".

Line 454: Check if this reference is correct, at the end appears an abbreviation "*edn.*" which duplicates the info given before. Maybe it just depends on the reference organizer you adopted.

Line 484: Please put "*scotland*" in capital letters "*Scotland*".

Line 534: Check if this reference is correct, at the end appears an abbreviation "*edn.*" which duplicates the info given before. Maybe it just depends on the reference organizer you adopted.

Line 563: Check if this reference is correct, at the end appears an abbreviation "*edn.*" which duplicates the info given before. Maybe it just depends on the reference organizer you adopted.

**Comments on figures and figure captions**

**Fig. 1**

In Fig.1a is reported "*scissor faults*" are you referring to a conjugate system of extensional faults with opposing dip direction? Can they be defined simply "*conjugate faults*"?

In Fig.1a there is a white rectangle in the background of the label "*scissor faults*" which partially overlaps with the outcrop image. Please shift upward the white rectangle to avoid intersection.

In the legend in Fig.1e you indicate "*current ripples*", but as you wrote for "*plant fragment and intraclast*" you should use the singular form instead of the plural.

In the caption you state "*scissor faults*", see if you prefer to use the term "*conjugate faults*" if you feel it could be more correct.

In the caption is reported "*rip-up clasts*", here you can add the nature of the eroded clasts "*pelitic rip-up clasts*".

In the caption please correct "*course sand*" with "*coarse sand*".

**Fig. 2**

In the figure caption please correct "*dashes lines*" with "*dashed lines*".

**Fig. 3**

In the figure caption check if you want to keep the lower case version "*p- and s-waves*" or the capitalized one "*P- and S-waves*".

In the figure caption you write "*x-ray*", while in the rest of the manuscript is written with capital letters "*X-ray*". Consider revising this point to keep the same style throughout all the text.

**Fig. 4**

In the caption please erase one of the highlighted "*the*" the avoid repetition.

**Fig. 6**

In the caption please evaluate to change "*major parts*" with "*the vast majority of...*".

In the caption please add a hyphen between "*intergranular*" to "*inter-granular*".

**Fig. 7**

In the figure caption check if you want to keep the lower case version "*p- and s-waves*" or the capitalized one "*P- and S-waves*".

**Fig. 8**

In the figure caption check if you want to keep the lower case version "*p- and s-waves*" or the capitalized one "*P- and S-waves*".

**Fig. 9**

To ease the distinction of the different lines you can try to enlarge a bit both the graphs, or you may assign different colors for every face of the cube.

**Table 1**

In the table caption check if you want to keep the lower case version "*p- and s-waves*" or the capitalized one "*P- and S-waves*".

[revised manuscript text omitted]

---

## Referee Comment (RC2) · Giacomo Medici (Referee) · 28 May 2020

**Solid Earth, Review**

**High-Resolution Analysis of the Physicochemical Characteristics of Sandstone Media at the Lithofacies Scale**

The potential paper is well written, valid and original. Indeed, models that combine spatial representation of physical and chemical parameters contrast large parts of literature on sandstone aquifers and reservoirs that focus on facies and permeability upscaling.

I strongly support the publication of this paper in *Solid Earth.* However, revisions and reply to general and specific comments are needed and there is my availability to review the paper a second time in case of request of the editors.

General comments

The authors should consider literature on the physiochemical properties of sandstone media more widely in the introduction and discussion. I recognize that the paper is original, but the authors should clarify better the reason in the introduction. As expressed above, models that combine spatial representation of chemical and physical parameters contrasts large part of literature on sandstone that largely focuses on 3D representation of the physical properties in three dimensions. Other papers exclusively treat the chemical properties of sandstone aquifers. Although the paper is generally well written I can see problems in the organization of the conclusions.

Please, refer to the comments below that aim to support resolution of problems and bring the impact out of your research.

Specific comments

1.0 Introduction

Lines 29-30 Add papers that treat upscaling and spatial properties of sandstone with regards to permeability issues related to nuclear waste repositories and hydrocarbon reservoirs.

- Kiryukhin, A.V., Kaymin, E.P. and Zakharova, E.V., 2008. Using TOUGHREACT to model laboratory tests on the interaction of NaNO3-NaOH fluids with sandstone rock at a deep radionuclide repository site. *Nuclear technology, 164*(2), pp.196-206.

- Medici, G., West, L.J. and Mountney, N.P., 2016. Characterizing flow pathways in a sandstone aquifer: tectonic vs sedimentary heterogeneities. *Journal of contaminant hydrology*, *194*, pp.36-58.

- Medici, G., West, L.J., Mountney, N.P. and Welch, M., 2019. Permeability of rock discontinuities and faults in the Triassic Sherwood Sandstone Group (UK): insights for management of fluvio-aeolian aquifers worldwide. *Hydrogeology Journal*, *27*(8), pp.2835-2855.

Lines 32-34 Again, I suggest updated literature on the topic for low porosity layers that reduce flow at the scale of the pumping tests in sandstone.

- Hamdi, Hamidreza, Philippe Ruelland, Pierre Bergey, and Patrick WM Corbett. "Using geological well testing for improving the selection of appropriate reservoir models." *Petroleum Geoscience* 20, no. 4 (2014): 353-368.

- Medici, G., West, L.J. and Mountney, N.P., 2019. Sedimentary flow heterogeneities in the Triassic UK Sherwood Sandstone Group: Insights for hydrocarbon exploration. *Geological Journal*, *54*(3), pp.1361-1378.

- Jackson, M.D., Muggeridge, A.H., Yoshida, S. and Johnson, H.D., 2003. Upscaling permeability measurements within complex heterolithic tidal sandstones. *Mathematical Geology*, *35*(5), pp.499-520.

- Tellam, J.H. and Barker, R.D., 2006. Towards prediction of saturated-zone pollutant movement in groundwaters in fractured permeable-matrix aquifers: the case of the UK Permo-Triassic sandstones. *Geological Society, London, Special Publications*, *263*(1), pp.1-48.

- Tidwell, V.C. and Wilson, J.L., 1997. Laboratory method for investigating permeability upscaling. *Water Resources Research*, *33*(7), pp.1607-1616.

Lines 25-68 Overall very good introduction. I suggest to add two or three sentences to explain not only which is your observation scale but also where it lies. Your outputs lie between the core plug and pumping test scale. Hence, your research contributes to bridge the gap between the two scales. See below relevant publications on the upscaling properties of sandstone aquifers/reservoirs.

- Corbett, P.W., Hamdi, H. and Gurav, H., 2012. Layered fluvial reservoirs with internal fluid cross flow: a well-connected family of well test pressure transient responses. *Petroleum Geoscience*, *18*(2), pp.219-229.

- Medici, G., West, L.J. and Mountney, N.P., 2018. Characterization of a fluvial aquifer at a range of depths and scales: the Triassic St Bees Sandstone Formation, Cumbria, UK. *Hydrogeology journal*, *26*(2), pp.565-591.

- Zheng, S.Y., Corbett, P.W., Ryseth, A. and Stewart, G., 2000. Uncertainty in well test and core permeability analysis: a case study in fluvial channel reservoirs, northern North Sea, Norway. *AAPG bulletin*, *84*(12), pp.1929-1954.

2. Measurement campaign

Line 109 "Hassler cell permeameter". I understand that you provide a reference. But, I think the manuscript would benefit of a sentence that explains the basic principal of your permeameter.

Lines 182-183 I leave to the authors the decision to state typical ranges of flow anisotropies ($K_h/K_v$) at the centimetre-meter scale in sandstones providing general references. Typical flow anisotropies are ~10-500 in sandstone aquifers with lower value in channalized sandstone of fluvial and deltaic origin.

3. Results

Lines 238-239 I suggest described by Fongngern et al. (2018).

Lines 306-307 Possible adding a short explanation on the reason why inverse distance and kriging provide comparable results? I guess the geometry that needs to be interpolated is relatively simple.

Lines 316-323 Realistic values of intrinsic permeability but very low. Please, justify your outputs with reference to the rock-type/lithofacies. The reason of this low permeability should be the sheet-like sandstone nature of the geological material tested. It's well known that sheet like sandstone are not very conductive for the fluids.

I'm inviting the author to make more evident in the paper the relation between sedimentology and intrinsic permeability.

4. Discussion

Line 379 If the authors want to enlarge bibliography on sandstone mineralogy and diagenesis. I suggest the following papers:

- Ixer, R.A., Turner, P. and Waugh, B., 1979. Authigenic iron and titanium oxides in Triassic red beds:(St. Bees Sandstone), Cumbria, northern England. *Geological Journal*, *14*(2), pp.179-192.

- Van Keer, I., Muchez, P.H. and Viaene, W., 1998. Clay mineralogical variations and evolutions in sandstone sequences near a coal seam and shales in the Westphalian of the Campine Basin (NE Belgium). *Clay Minerals, 33*(1), pp.159-169.

Line 380 I invite the authors to avoid the use of "because" in a scientific paper. Aside from minor issues the manuscript is very well written.

5 Conclusions

Lines 400-401 I agree on the use of bulletin points. I suggest adding one or two sentences to introduce your four points. This passage from standard text to bulletin points sound chunky to the readers.

Lines 417-419 Future work is introduced here in an abrupt way. Also, better avoiding new topics in the conclusions. It's fine to introduce future research scenarios. But, in this case, the topic needs to be analysed in the discussion section.

Figures and tables

All figures and tables of publishable quality. I remind the authors to comment on the low intrinsic permeabilities (see Fig. 14) of the studied deposits.

Fig. 6 Make this image larger.

Fig. 7 Figures on axes larger.

Overall, very good contribution to the petro-hydraulic properties of porous sandstone.

My best wishes,

Giacomo Medici

---

## Author Comment (AC1) · 4 Jun 2020

**Response to the comments of reviewer #1**

**Manuscript se-2020-13, Linsel et al.**

**"High-Resolution Analysis of the Physicochemical Characteristics of Sandstone Media at the Lithofacies Scale"**

Dear Reviewers and Editor,
we would like to express our sincerest thanks to the reviewers who both provided a very constructive feedback, which helped to significantly enhance the quality of our manuscript. Below please find a point-by-point response to the general and specific comments of Mattia Pizzati (reviewer #1). The response is provided in blue color whereas replaced and new text in the manuscript is indicated by *italic blue* font.

**General comments**
Dear Authors and Editor,

below you can find the review of the submitted manuscript.
Revisions are made by describing the issues found line by line and also on the text file of the manuscript, in which critical points were highlighted in green color. Comments on figures, figure captions and tables are presented at the end of this file as well.

This manuscript is focused on the analysis of physical, chemical petrophysical and diagenetic properties characterizing a fluvial-lacustrine sandstone mouth bar at the sedimentological facies scale. Applied methods include both field and laboratory measurements, statistical validation and cross-checking.

Laboratory analyses and statistical tests are thoroughly described and explained, and even if the section describing methods may appear quite long compared to the rest of the manuscript, I feel it is necessary to the full comprehension of the results and discussion. This is particularly true for readers who may not have a strong statistical background and without this support they could be discouraged in going through the text.

The quality of written English is very high, sentences are clear and reading is fluent. I found some minor mistakes and added a few suggestions to further improve the structure of sentences.

Figures are schematic, clear and of good quality. I inserted a few suggestions aimed to improve their visibility and minor corrections to be done mainly on legends and captions.

Data presented fully support the thesis the authors wanted to discuss. This manuscript represents a good example of how important is the detailed study of petrophysical, diagenetic and physical properties of sandstone at the scale of the sedimentary facies.

Maybe it would have been nice to have an additional paragraph inside the discussion in which the authors could have described the implications for water-oil reservoir exploitation and maybe also the up-scaling of the properties they discuss. However, this could be much beyond the original aim of the present work, but it may represent a cue for future studies.

Thank you for your thoughtful and constructive comments. Indeed, the implications for the water-oil reservoir exploitation and for other processes connected to subsurface utilization such as mining, geothermal heat extraction or nuclear waste disposal are quite impacting. In a follow-up paper, we describe, how the local geological variability, as described in this manuscript, can be accounted for in spatial property predictions using sequential simulations. In the scope of this manuscript, we have added another conclusion as follows:

"…
*5. As shown in this study, the local geological variability should not be underestimated as an uncertainty factor in spatial predictions and up-scaling procedures. In fact, the local geological variability of physicochemical properties might nearly cover the variability being present in an entire formation. Therefore, a high-resolution analysis of physicochemical rock properties can assist in assessing the uncertainty of field-scale property models, which is induced by the local geological variability at the lithofacies scale.*
…"

Another point that could be of real interest is to test the differences or similarities between the properties of undeformed sandstone and fault rocks, since in the described outcrop at least two faults with few meter of displacement are present. Again consider this as a hint for next research topics.

As the study was not aimed at estimating the influence of fault-compartmentalization and accompanying rock deformation onto the petrophysical properties of rock, we did not take that into account here. We are, however, grateful for the suggestion to consider that in a future research project.

I think this paper is worth to be accepted on Solid Earth after the completion of technical minor revisions.

I really hope this review will be helpful to the improvement of the final version of your manuscript. It is a nice research topic and deserves to be developed in the best way.
Do not hesitate to contact me in case any questions or doubts arise from these comments:

mattia.pizzati@studenti.unipr.it

**Specific comments**

Line 14: Please add a hyphen between "*intergranular*" to "*inter-granular*".

We have added a hyphen at each occurrence of "intergranular" in the running text.

Line 19: Perhaps this is just a tiny detail, but also in textbooks I always read "P and Swaves" with capital letters. In the entire text of the manuscript it is reported with lower case. Consider

if this suggestion suits you. To ease the identification, I underlined "p and s-waves" throughout all the text.

Yes, we agree that P- and S-wave is provided with capital letters in most scientific works and we revised the expression in the text.

Line 20: Here maybe "*strictly*" may sound better than "*inevitably*".

We agree and revised the expression.

Line 22: Consider substituting "*In fact*" with "*Following this,*".

We agree and revised the text accordingly.

Line 50: Please add a hyphen in "*interrelationships*" to "*inter-relationships*".

We have added a hyphen at each occurrence of "interrelationships" in the running text.

Line 54: Please change "*are taken*" with "*have been sampled*".

We have revised the given passage in the text.

Line 56: Perhaps the term "*series*" sounds a bit too generic. Is it possible to adopt the word "*orogeny*" or "*tectonic event*"? I have limited knowledge of regional geology of Germany, so evaluate if this term need to be corrected or not.

The Cisularian is the first series of the Permian. Therefore, we would prefer to stick with our original expression here.

Line 59: Check if you want to keep the lower case version "*p- and s-waves*" or the capitalized one "*P- and S-waves*".

Revision has been implemented as described above.

Line 72: When you are referring to specific hierarchical stratigraphic nomenclature (formation, group and so on) you should adopt the capital letters. Following this, please correct " *Disibodenberg formation* " with "*Disibodenberg Formation*".

Yes, that is true. We revised it accordingly.

Line 73: You provided the present-day depth of the Disibodenberg Formation in the Upper Rhine Graben. Is it possible to constrain, even roughly, the maximum burial depth experienced by the studied sandstone in your field site? It could be very interesting since further below you describe the effects of mechanical and chemical compaction affecting the sandstone.

Thank you very much for this important comment. We have inserted a sentence which refers to the study of Henk (1992) whose results indicate that the maximum thickness of the

overburden in this area of the Saar-Nahe-Basin was in the range between 1,950 m and 2400 m.

"*The maximum past overburden of the field site can be estimated to be between 1,950 m and 2,400 m as indicated by shale-compaction analyses, which were performed by Henk (1992).*"

Line 76: To avoid the repetition of the word "*formation*" you can substitute the second highlighted word simply with "*it*".

We agree.

Line 77: Here maybe "*were sampled*" or "*were extracted*" sounds better than "*were taken*".

We agree and changed the expression into *"were extracted"*

Line 81: Please change the sequence of these words from "*The cubes both were...*" to "*The cubes were both...*".

Yes, we reordered the words.

Line 84: With the term "*granularity*" are you describing the angularity of grains composing the mouth-bar sandstone strata? If so I believe that "*angularity*" provides a clearer and more straightforward idea of what you are describing.

We agree and changed "granularity" to "*angularity*".

Lines 86-87: Since you are describing a methodology with only one bibliography reference maybe you should turn to the singular form "*This approach, however, does often...*".

Yes, thank you for this suggestion!

Line 88: Instead of "*we measured the faces of the cubes...*" if you prefer adopt the form "*we performed analyses on faces of the cubes...*".

Yes, that sounds better.

Line 96: Can you define the size of the elementary cell used in 3D models?

We have added a sentence, in which the size of an elementary cell is provided

"*The elementary cell of OSB1_c has a volume of $5.7 \cdot 10^{-7}$ $m^3$ whereas OSB2_c's elementary cells have a volume of $3 \cdot 10^{-7}$ $m^3$.*"

Line 97: Perhaps the title of the paragraphs could be changed to "*In situ measurements*" or "*Field measurements*".

As we are describing the laboratory experiments in this section, we have changed the title to "*Laboratory experiments*" accordingly. We hope that our suggestion might suit you well too.

Line 102: Check if you want to keep the lower case version "*p- and s-waves*" or the capitalized one "*P- and S-waves*".

Revision has been implemented as described above.

Line 105: Check if you want to keep the lower case version "*p- and s-waves*" or the capitalized one "*P- and S-waves*".

Revision has been implemented as described above.

Line 111: Since in the same sentence you used the past simple, I would recommend to keep the same style. If you agree change "*do*" with "*did*".

Yes, we agree and revised it.

Line 117: Check if you want to keep the lower case version "*p- and s-waves*" or the capitalized one "*P- and S-waves*".

Revision has been implemented as described above.

Line 119: Here maybe erase the second "*travel*" since it could be a repetition.

We needed to rearrange this part of the text as the first occurrence of "travel" does not refer to "density". The new sentence sounds like follows

"*The wave velocity is a function of travel length and time together with the density of the material*"

Line 119: Check if you want to keep the lower case version "*p- and s-waves*" or the capitalized one "*P- and S-waves*".

Revision has been implemented as described above.

Line 129: Perhaps "*insights on*" sounds better than "*insight into*".

To the best of our knowledge, the usual preposition for "insight" is "into". Therefore, we would prefer to stick with our original expression here.

Line 130: I would rework the sentence as indicated: "*can significantly impact the petrophysical properties*".

We have reordered the sentence accordingly.

Line 151: Check if here you want to use "*neighborhood*" or "*neighbor*".

We would prefer to stick with our formulation here as we are in fact considering a neighborhood instead of a single neighbor here.

Line 186: Check if you want to keep the lower case version "*p- and s-waves*" or the

capitalized one "*P- and S-waves*".

Revision has been implemented as described above.

Lines 191-192: Check if you want to keep the lower case version "*p- and s-waves*" or the capitalized one "*P- and S-waves*".

Revision has been implemented as described above.

Line 244: Consider if "*sampled*" suits better than "*taken*".

Yes, it does, and we changed it accordingly.

Line 247: In the previous sentence you state that the basal part of the bed can be classified as Bauma A subdivision, which ideally implies massive medium-coarse sandstone. However, in this line it is reported the presence of "*sub-horizontal layering*", that may contrast with what is described above (massive should mean structureless). Can you please clarify this point? I fully understand that the Bauma subdivision is something purely ideal, and differences from the model may occur. Are these "*sub-horizontal layers*" similar to crude laminations, which typically occur in the upper part of the Bauma A subdivision?

This is a very good point, which may not have been clarified appropriately in the text. In a Bouma A interval the rip-up clasts may build a form of layering above the base if they are exposed to a decent degree of buoyancy during transport. In order to clarify this issue, we now refer to the schematic provided by Middleton (1993) in which OSB1_c corresponds to Bouma A and OSB2_c corresponds to Bouma E. The text in the final revised version has been adapted accordingly.

Line 247: By "*homogeneous*" here you mean massive, without any structure or are you referring to the grain size. Sorry for being blunt here, but personally "*homogeneous*" fells a bit to generic.

We implemented your suggestion here. Thank you!

Line 251: Here maybe instead of "*transitions*" you can use "*decreases*".

We used your suggested formulation here.

Line 252: See if this reworking suits you: "*Likewise, sorting increases from poor to moderate*".

Sound way better, thank you!

Line 252: Instead of "*continuously*" perhaps use "*throughout the entire sample volume*".

Perfect, we adapted the text accordingly.

Line 255: To be more explicit please add "... *sub-vertically with respect to bedding*".

Yes, your suggestion makes the expression now very explicit. Thanks for that!

Line 256-257: Could you please separate the percentage of feldspar and the one pertaining to the lithic fragments (mica, rock fragments)? Just to have the percentage relative to quartz, feldspar and rock fragments.

Yes, we clarified it in the text as the 20-30% was only meant to represent the fraction of feldspar rather than the lithic fragments.

"*The original rigid detrital components consist of 50–60% quartz, 20–30% strongly altered feldspar as well as 10–25% lithic fragments.*"

Line 258: Please define if possible the nature of "*ductile, autochthonous grains*".

We have added "*pelite*" before "grains". Hopefully, the nature of the grains can be clarified by this additional information.

Line 260: Here I believe you can use the in-text citation reference "*Herron (1988)*" instead of the form in parentheses.

Yes indeed. Thank you!

Line 261: Here I would erase "*produced*".

"produced" has been erased.

Line 263: You can add "*primary*" before "*inter-granular*" since here you are referring to the original porosity of the sandstone prior to compaction.

Yes, we agree and added "*primary*" before "inter-granular".

Line 272: You can use the word "*analyses*" to avoid repetition of "*measurements*".

Yes, we revised the expression.

Line 279: Check if you want to keep the lower case version "*p- and s-waves*" or the capitalized one "*P- and S-waves*".

Revision has been implemented as described above.

Line 282: Check if you want to keep the lower case version "*p- and s-waves*" or the capitalized one "*P- and S-waves*".

Revision has been implemented as described above.

Line 309: Check if you want to keep the lower case version "*p- and s-waves*" or the capitalized one "*P- and S-waves*".

Revision has been implemented as described above.

Line 310: Please correct the misspelled word "*petrophyscial*" with "*petrophysical*".

We corrected the typo. Thanks for the hint.

Line 326: Here I would erase "*The*" to make the sentence lighter.

Thank you for the suggestion but we would rather keep the "The" at the beginning of the sentence and leave it up to the typesetters whether it can be erased or not.

Line 326: Maybe "*that*" is more correct than "*what*".

Indeed, we revised it.

Line 368: Here you should use the plural form "*lead*" instead of the third person "*leads*", since the subject of the sentence is plural "*alteration products*".

Yes, we agree and revised it.

Line 388: Please add a hyphen between "*intergranular*" to "*inter-granular*".

We added a hyphen at each occurrence of "intergranular" in the running text.

Line 390: See if this sentence reworking sounds: "*...which might have further enforced degradation of porosity and permeability*".

This sounds way better. Thank you!

Line 391: Maybe you can change "*analogies*" simply with "*ones*".

Yes, we agree. Also, we have split up the sentence into two ones and added "matrix-rich" in front of "media" in order to emphasize the distinction between the different types of porous media.

„*Primary matrix typically plugs the pore throats of porous, matrix-rich media. This reduces the ability to conduct fluids compared to matrix-free ones.*"

Line 409: Please add a hyphen between "*intergranular*" to "*inter-granular*".

We added a hyphen at each occurrence of "intergranular" in the running text.

Line 454: Check if this reference is correct, at the end appears an abbreviation "*edn.*" which duplicates the info given before. Maybe it just depends on the reference organizer you adopted.

Yes, this was indeed an inconsistency in the reference organizer. Thank you for the remark.

Line 484: Please put "*scotland*" in capital letters "*Scotland*".

Corrected.

Line 534: Check if this reference is correct, at the end appears an abbreviation "*edn.*" which duplicates the info given before. Maybe it just depends on the reference organizer you adopted.

Yes, this was indeed an inconsistency in the reference organizer. Thank you for the remark.

Line 563: Check if this reference is correct, at the end appears an abbreviation "*edn.*" which duplicates the info given before. Maybe it just depends on the reference organizer you adopted.

Yes, this was indeed an inconsistency in the reference organizer. Thank you for the remark.

**Fig. 1**

In Fig.1a is reported "*scissor faults*" are you referring to a conjugate system of extensional faults with opposing dip direction? Can they be defined simply "*conjugate faults*"?

The fault belongs to the Lauter fault zone which constitutes an anastomosing, steeply dipping set of strike-slip transform faults with interleaved lens-shaped segments. Furthermore, the lens-shaped segments are characterized by block rotations the axis of which is oblique to the general fault trend (Stoffhofen, 1998). We changed "scissor faults" to the more generic "transform faults" and placed a reference to the article of Stollhofen (1998).

"*The outcrop is compartmentalized in the central part by two transform faults, which belong to the Lauter fault zone (Stollhofen, 1998). The strike-slip faults provide offsets of a few meters.*"

In Fig.1a there is a white rectangle in the background of the label "*scissor faults*" which partially overlaps with the outcrop image. Please shift upward the white rectangle to avoid intersection.

We removed the background of the label.

In the legend in Fig.1e you indicate "*current ripples*", but as you wrote for "*plant fragment and intraclast*" you should use the singular form instead of the plural.
In the caption you state "*scissor faults*", see if you prefer to use the term "*conjugate faults*" if you feel it could be more correct.

We agree and adapted the text.

In the caption is reported "*rip-up clasts*", here you can add the nature of the eroded clasts "*pelitic rip-up clasts*".

We agree and adapted the text.

In the caption please correct "*course sand*" with "*coarse sand*".

Of course, that is such an obvious typo. Thank you very much!

**Fig. 2**

In the figure caption please correct "*dashes lines*" with "*dashed lines*".

Again, a very obvious typo. Thanks for the remark.

**Fig. 3**

In the figure caption check if you want to keep the lower case version "*p- and s-waves*" or the capitalized one "*P- and S-waves*".

Revision has been implemented as described above.

In the figure caption you write "*x-ray*", while in the rest of the manuscript is written with capital letters "*X-ray*". Consider revising this point to keep the same style throughout all the text.

We revised it.

**Fig. 4**

In the caption please erase one of the highlighted "*the*" the avoid repetition.

We revised it.

**Fig. 6**

In the caption please evaluate to change "*major parts*" with "*the vast majority of...*".

Thanks for the suggestion. We revised it accordingly.

In the caption please add a hyphen between "*intergranular*" to "*inter-granular*".

We revised it.

**Fig. 7**
In the figure caption check if you want to keep the lower case version "*p- and s-waves*" or the capitalized one "*P- and S-waves*".
Revision has been implemented as described above.

**Fig. 8**

In the figure caption check if you want to keep the lower case version "*p- and s-waves*" or the capitalized one "*P- and S-waves*".

Revision has been implemented as described above.

**Fig. 9**

To ease the distinction of the different lines you can try to enlarge a bit both the graphs, or you may assign different colors for every face of the cube.

We both rearranged and resized the elements of the figure and assigned different colors for every face of the cubes.

**Table 1**

In the table caption check if you want to keep the lower case version "*p- and s-waves*" or the capitalized one "*P- and S-waves*".

Revision has been implemented as described above.

---

## Author Comment (AC2) · 4 Jun 2020

**Response to the comments of reviewer #2**

**Manuscript se-2020-13, Linsel et al.**

**"High-Resolution Analysis of the Physicochemical Characteristics of Sandstone Media at the Lithofacies Scale"**

Dear Reviewers and Editor,
we would like to express our sincerest thanks to the reviewers who both provided a very constructive feedback, which helped to significantly enhance the quality of our manuscript. Below please find a point-by-point response to the general and specific comments of Giacomo Medici (reviewer #2). The response is provided in blue color whereas replaced and new text in the manuscript is indicated by *italic blue* font.

**General comments**
The authors should consider literature on the physiochemical properties of sandstone media more widely in the introduction and discussion. I recognize that the paper is original, but the authors should clarify better the reason in the introduction. As expressed above, models that combine spatial representation of chemical and physical parameters contrasts large part of literature on sandstone that largely focuses on 3D representation of the physical properties in three dimensions. Other papers exclusively treat the chemical properties of sandstone aquifers. Although the paper is generally well written I can see problems in the organization of the conclusions. Please, refer to the comments below that aim to support resolution of problems and bring the impact out of your research.

We would like to thank Giacomo Medici for the thorough review and refer to the point-by-point response of the specific comments as the general comments are addressed there. Generally, we have added more literature to the Introduction, Discussion and Conclusion, revised some minor technical issues and rearranged the Conclusions as suggested by the reviewer.

**Specific comments**

1.0 Introduction

Lines 29-30 Add papers that treat upscaling and spatial properties of sandstone with regards to permeability issues related to nuclear waste repositories and hydrocarbon reservoirs.

- Kiryukhin, A.V., Kaymin, E.P. and Zakharova, E.V., 2008. Using TOUGHREACT to model laboratory tests on the interaction of NaNO3-NaOH fluids with sandstone rock at a deep radionuclide repository site. Nuclear technology, 164(2), pp.196-206.

- Medici, G., West, L.J. and Mountney, N.P., 2016. Characterizing flow pathways in a sandstone aquifer: tectonic vs sedimentary heterogeneities. Journal of contaminant hydrology, 194, pp.36-58.

- Medici, G., West, L.J., Mountney, N.P. and Welch, M., 2019. Permeability of rock discontinuities and faults in the Triassic Sherwood Sandstone Group (UK): insights for management of fluvio-aeolian aquifers worldwide. Hydrogeology Journal, 27(8), pp.2835-2855.

We are grateful to the reviewer for providing us with these valuable references. We have placed them into our running text as you recommended.

Lines 32-34 Again, I suggest updated literature on the topic for low porosity layers that reduce flow at the scale of the pumping tests in sandstone.

- Hamdi, Hamidreza, Philippe Ruelland, Pierre Bergey, and Patrick WM Corbett. "Using geological well testing for improving the selection of appropriate reservoir models." Petroleum Geoscience 20, no. 4 (2014): 353-368.

- Medici, G., West, L.J. and Mountney, N.P., 2019. Sedimentary flow heterogeneities in the Triassic UK Sherwood Sandstone Group: Insights for hydrocarbon exploration. Geological Journal, 54(3), pp.1361-1378.

- Jackson, M.D., Muggeridge, A.H., Yoshida, S. and Johnson, H.D., 2003. Upscaling permeability measurements within complex heterolithic tidal sandstones. Mathematical Geology, 35(5), pp.499-520.

- Tellam, J.H. and Barker, R.D., 2006. Towards prediction of saturated-zone pollutant movement in groundwaters in fractured permeable-matrix aquifers: the case of the UK Permo-Triassic sandstones. Geological Society, London, Special Publications, 263(1), pp.1-48.

- Tidwell, V.C. and Wilson, J.L., 1997. Laboratory method for investigating permeability upscaling. Water Resources Research, 33(7), pp.1607-1616.

We have also incorporated these suggested references into our introduction. Thanks again!

Lines 25-68 Overall very good introduction. I suggest to add two or three sentences to explain not only which is your observation scale but also where it lies. Your outputs lie between the core plug and pumping test scale. Hence, your research contributes to bridge the gap between the two scales. See below relevant publications on the upscaling properties of sandstone aquifers/reservoirs.

- Corbett, P.W., Hamdi, H. and Gurav, H., 2012. Layered fluvial reservoirs with internal fluid cross flow: a well-connected family of well test pressure transient responses. Petroleum Geoscience, 18(2), pp.219-229.

- Medici, G., West, L.J. and Mountney, N.P., 2018. Characterization of a fluvial aquifer at a range of depths and scales: the Triassic St Bees Sandstone Formation, Cumbria, UK. Hydrogeology journal, 26(2), pp.565-591.

- Zheng, S.Y., Corbett, P.W., Ryseth, A. and Stewart, G., 2000. Uncertainty in well test and core permeability analysis: a case study in fluvial channel reservoirs, northern North Sea, Norway. AAPG bulletin, 84(12), pp.1929-1954.

Again, thank you very much for providing relevant literature references. We have added the references to the running text and provided a paragraph at the end of the introduction which aims at describing the scale of investigations and which role it plays in upscaling procedures.

"*The research outputs of this study lie between the scale of a core plug measurement and a wireline log/pumping test (Medici et al., 2018). Hence, we aim to contribute towards estimating the uncertainty that must be accounted for when performing up- or down-scaling between those two scales of investigation (Zheng et al., 2000; Jackson et al., 2003; Corbett et al., 2012; Hamdi et al., 2014).*"

2. Measurement campaign

Line 109 "Hassler cell permeameter". I understand that you provide a reference. But, I think the manuscript would benefit of a sentence that explains the basic principal of your permeameter.

We have added a sentence on the basic principle of the permeameter as follows:

"*The Hassler cell is a gas-driven permeameter which measures the permeability of a cylinder-shaped rock sample under steady-state gas flow.*"

Lines 182-183 I leave to the authors the decision to state typical ranges of flow anisotropies (Kh/Kv) at the centimetre-meter scale in sandstones providing general references. Typical flow anisotropies are ~10-500 in sandstone aquifers with lower value in channalized sandstone of fluvial and deltaic origin.

We have inserted a sentence about the typical range of kv to kh:

"*The intrinsic permeability, for example, provides typical ranges for the ratio between the vertical ($k_v$) and horizontal permeability ($k_h$) of $10^{-5}$ to 1 (Ringrose and Bentley, 2015).*"

Also, we have added a comment on the observed anisotropy of the intrinsic permeability in line 277:

"*Also, the intrinsic permeability does not show a significant anisotropy.*"

3. Results

Lines 238-239 I suggest described by Fongngern et al. (2018).

Yes, thank you for the remark. We revised it in the text.

Lines 306-307 Possible adding a short explanation on the reason why inverse distance and kriging provide comparable results? I guess the geometry that needs to be interpolated is relatively simple.

Both interpolation procedures are so-called exact interpolators, which means that at each known point, the interpolation function takes the value of that exact point. Due to the high sampling density, the patterns are thus similar. Following that, we adapted the text here like: "…*provide comparable patterns, which is due to the high sampling density*."

Lines 316-323 Realistic values of intrinsic permeability but very low. Please, justify your outputs with reference to the rock-type/lithofacies. The reason of this low permeability should be the sheet-like sandstone nature of the geological material tested. It's well known that sheet like sandstone are not very conductive for the fluids. I'm inviting the author to make more evident in the paper the relation between sedimentology and intrinsic permeability.

This is an important comment, which we tried to resolve by adding two sentences in the discussion of the relationship between porosity and permeability as follows:

"… *In addition, these observations are well reflected by the very low values of the intrinsic permeability in both rock cubes. Another reason for the very low intrinsic permeability is the high amount of primary clay and the low maturity of deltaic sheet-like distributary mouth bar deposits (Tye and Hickey, 2001).*"

4. Discussion

Line 379 If the authors want to enlarge bibliography on sandstone mineralogy and diagenesis. I suggest the following papers:

- Ixer, R.A., Turner, P. and Waugh, B., 1979. Authigenic iron and titanium oxides in Triassic red beds:(St. Bees Sandstone), Cumbria, northern England. Geological Journal, 14(2), pp.179-192.

- Van Keer, I., Muchez, P.H. and Viaene, W., 1998. Clay mineralogical variations and evolutions in sandstone sequences near a coal seam and shales in the Westphalian of the Campine Basin (NE Belgium). Clay Minerals, 33(1), pp.159-169.

Thank you so much for these valuable references. We have considered them in our manuscript.

Line 380 I invite the authors to avoid the use of "because" in a scientific paper. Aside from minor issues the manuscript is very well written.

Thank you for the suggestion. We substituted this word throughout the manuscript.

5 Conclusions

Lines 400-401 I agree on the use of bulletin points. I suggest adding one or two sentences to introduce your four points. This passage from standard text to bulletin points sound chunky to the readers.

*We have inserted an introductory sentence before the conclusions:*

"*All in all, the following conclusions can be drawn from this study:*"

Lines 417-419 Future work is introduced here in an abrupt way. Also, better avoiding new topics in the conclusions. It's fine to introduce future research scenarios. But, in this case, the topic needs to be analysed in the discussion section.

*We understand your comment and removed the outlook from the conclusions accordingly.*

Figures and tables All figures and tables of publishable quality. I remind the authors to comment on the low intrinsic permeabilities (see Fig. 14) of the studied deposits.

*We would like to mention that we commented on the low intrinsic permeabilities in the Discussion section as outlined for the comment on Lines 316-323.*

Fig. 6 Make this image larger.

*We increased the size of the image.*

Fig. 7 Figures on axes larger.

*The figures have been adapted.*